# Six Global Biomass Burning Emission Datasets: Inter-comparison and Application in one Global Aerosol Model

Xiaohua Pan[1,2,*] Charles Ichoku[3], Mian Chin[2], Huisheng Bian[4,2], Anton Darmenov[2], Peter Colarco[2], Luke Ellison[5,2], Tom Kucsera[6,2], Arlindo da Silva[2], Jun Wang[7], Tomohiro Oda[6,2], Ge Cui[7]

1. Earth System Science Interdisciplinary Center, University of Maryland, College Park, MD, USA
2. NASA Goddard Space Flight Center, Greenbelt, MD, USA
3. Howard University, Washington DC, USA
4. Joint Center for Earth Systems Technology, University of Maryland Baltimore City, Baltimore, MD, USA
5. Science Systems and Applications, Inc., Lanham, MD, USA
6. Universities Space Research Association, Columbia, MD, USA
7. University of Iowa, College of Engineering, Iowa City, IA, USA

* xpan333@umd.edu

## Abstract

Aerosols from biomass burning (BB) emissions are poorly constrained in global and regional models, resulting in a high level of uncertainty in understanding their impacts. In this study, we compared six BB aerosol emission datasets for 2008 globally as well as in 14 regions. The six BB emission datasets are: (1) GFED3.1 (Global Fire Emissions Database version 3.1); (2) GFED4s (GFED version 4 with small fires); (3) FINN1.5 (FIre INventory from NCAR version 1.5); (4) GFAS1.2 (Global Fire Assimilation System version 1.2); (5) FEER1.0 (Fire Energetics and Emissions Research version 1.0), and (6) QFED2.4 (Quick Fire Emissions Dataset version 2.4). The global total emission amounts from these six BB emission datasets differed by a factor of 3.8, ranging from 13.76 to 51.93 Tg for organic carbon and from 1.65 to 5.54 Tg for black carbon. In most of the regions, QFED2.4 and FEER1.0, which are based on satellite observations of fire radiative power (FRP) and constrained by aerosol optical depth (AOD) data from the Moderate Resolution Imaging Spectroradiometer (MODIS), yielded higher BB emissions than the rest by a factor of 2-4. By comparison, the BB emission estimated from GFED4s and GFED3.1, which are based on satellite burned-area data, without AOD constraints, were at the low end of the range. In order to examine the sensitivity of model simulated AOD to the different BB emission datasets, we ingested these six BB emission datasets separately into the same global model, the NASA Goddard Earth Observing System (GEOS) model, and compared the simulated AOD with observed AOD from the AErosol RObotic NETwork (AERONET) and the Multiangle Imaging SpectroRadiometer (MISR) in the 14 regions during 2008. In Southern hemisphere Africa (SHAF) and South America (SHSA), where aerosols tend to be clearly dominated by smoke in September, the simulated AOD were underestimated almost in all experiments compared to MISR, except for the QFED2.4 run in SHSA. The model-simulated AOD based on FEER1.0 and QFED2.4 were the closest to the corresponding AERONET data, being, respectively, about 73% and 100% of the AERONET observed AOD at Alta-Floresta in SHSA, and about 49% and 46% at Mongu in SHAF. The simulated AOD based on the other four BB emission datasets accounted for only ~ 50% of the AERONET AOD at Alta Floresta and ~ 20% at Mongu. Overall, during the biomass burning peak seasons, at most of the selected AERONET sites in each region,

the AOD simulated with QFED2.4 were the highest and closest to AERONET and MISR
observations, followed closely by FEER1.0. However, the QFED2.4 run tends to
overestimate AOD in the region of SHSA, and the QFED2.4 BB emission dataset is tuned
with the GEOS model. In contrast, the FEER1.0 BB emission dataset is derived in a more
model-independent fashion and is more physical-based since its emission coefficients are
independently derived at each grid box. Therefore, we recommend the FEER1.0 BB
emission dataset for aerosol-focused hindcast experiments in the two biomass-burning
dominated regions in the southern hemisphere, SHAF and SHSA (as well as in other
regions but with lower confidence). The differences between these six BB emission
datasets are attributable to the approaches and input data used to derive BB emissions, such
as whether AOD from satellite observations is used as a constraint, whether the approaches
to parameterize the fire activities are based on burned area, FRP, or active fire count, and
which set of emission factors is chosen.

## 1. Introduction

Biomass burning (BB) is estimated to contribute about 62% of the global particulate organic carbon (OC) and 27% of black carbon (BC) emissions annually (Wiedinmyer et al., 2011). Therefore, biomass burning emissions significantly affect air quality by acting as a major source of particulate matter (PM), and the climate system by modulating solar radiation and cloud properties. For instance, a number of studies have revealed that wildfire smoke exposure is harmful to human health by causing general respiratory morbidity and exacerbating asthma, because approximately 80–90% of the smoke particles produced by biomass burning fall within the $PM_{2.5}$ size range (PM with aerodynamic diameter less than 2.5 μm) (Reid et al., 2005, 2016). Moreover, biomass burning emissions have been shown to impact the atmospheric composition in different regions, such as South America (Reddington et al., 2016), Central America (Wang et al., 2006), sub-Saharan African region (Yang et al., 2013), Southeast Asia (Wang et al., 2013; Pan et al., 2018), Indo-China (Zhu et al., 2017), and Western Arctic (Bian et al., 2013). Additionally, BB-produced aerosols can also directly impact the upper troposphere and lower stratosphere via extreme pyro-convection events associated with intense wildfires that generate the storms injecting smoke particles and trace gases to high altitudes (e.g., Peterson et al., 2018). Therefore, emissions from biomass burning constitute a significant component of the climate system, and are crucial inputs required by chemical transport and atmospheric circulation models used to simulate the atmospheric compositions, radiation, and circulation processes involved in air-quality and climate-impact studies (e.g., van Marle et al., 2017).

With the advent of satellite remote sensing of active fire and burned areas in the last couple of decades, a number of global BB emission datasets based on these observations have become available (e.g., Ichoku et al., 2012). Six of such major BB datasets will be compared in this study, including three datasets based on burned area approaches, namely, the Fire INventory from NCAR (FINN, Wiedinmyer et al., 2011) and two versions of the Global Fire Emissions Database (GFED, van der Werf et al., 2006, 2010, 2017), and three datasets based on fire radiative power (FRP) approaches, namely, the Global Fire Assimilation System (GFAS, Kaiser et al., 2012) developed in the European Centre for Medium-Range Weather Forecasts (ECMWF), and two National Aeronautics and Space Administration (NASA) products, i.e., the Fire Energetics and Emissions Research algorithm (FEER, Ichoku and Ellison, 2014) and the Quick Fire Emissions Dataset (QFED, Darmenov and da Silva, 2015).

Although much progress has been made over the last couple of decades in improving the quality of BB emission datasets, for example, by incorporating more recent satellite measurements with better calibration and spatial resolution (e.g., van der Werf et al. 2010; 2017), biomass-burning aerosol emissions still have large uncertainty, and thus are still poorly constrained in models at global and regional levels (e.g., Liousse et al., 2010; Kaiser et al., 2012; Petrenko et al., 2012, 2017; Bond et al., 2013; Zhang et al., 2014; Pan et al., 2015; Ichoku et al., 2016a; Reddington et al., 2016; Pereira et al., 2016). Specifically, large uncertainty exists in the description of the magnitude, patterns, and drivers of wildfires and types of biomass burning (e.g., Hyer et al, 2011). For instance, a global enhancement of particulate matter BB emission by a factor of 3.4 was

recommended for GFAS by Kaiser et al. (2012) to match the corresponding observed
aerosol loading. Andreae (2019) commented that "In contrast to gaseous compounds,
which are chemically well defined, aerosols are complex and variable mixtures of organic
and inorganic species and comprise particles across a wide range of sizes. This affects in
particular the measurements of organic aerosol, black/elemental carbon, and size
fractionated aerosol mass".
A recent analysis with multiple models has been conducted under the auspices of the
Aerosol Comparisons between Observations and Models (AeroCom) Phase III biomass
burning emission experiments using the GFED version 3.1 (GFED3.1) emission dataset
as input to several models (hereinafter, "The AeroCom multi-model study",
https://wiki.met.no/aerocom/phase3-experiments) (M. Petrenko, personal
communication). The AeroCom Multi-model study concluded that the modelled aerosol
optical depth (AOD) from different models exhibits large diversity in most regions, i.e.
some models overestimate while other models underestimate. However, over two major
biomass burning dominated regions, South America and southern hemisphere Africa, all
models consistently underestimate AOD. That result suggests that the underestimation of
AOD in these two regions was more likely attributable to the GFED3.1 biomass burning
emission dataset rather than the model configurations.
Our study aims to explore multiple BB emission datasets, including GFED3.1, GFED
version 4 with small fires (GFED4s), FINN version 1.5, GFAS version 1.2, QFED
version 2.4, and FEER version 1.0, in order to investigate the discrepancies between
these six BB emission datasets by comparing them at both regional and global levels.
Such a comparative evaluation of BB emission datasets would show the differences
between them as well as how these differences propagate through the physical processes
of related aerosols in models, e.g., dry and wet deposition, transport, atmospheric
abundance, and the resulting AOD. Our study is expected to provide further insight into
the development of possible mitigation for the current large uncertainties in BB
emissions. Similar comparative studies of multiple BB aerosol emission datasets were
previously conducted at regional scales, e.g., by Zhang et al. (2014) in the northern sub-
Saharan African region, Pereira et al. (2016) and Reddington et al. (2019) in South
America, and Reddington et al. (2016) in the entire tropical region. The current study not
only provides a global assessment and analysis of these six BB emission datasets to
provide a world-wide perspective, but also examines their performance within 14 regions
(Fig. 1) which were previously defined for a series GFED-based studies (e.g., Van der
Werf et al., 2006, 2010, and 2017).
In the rest of this paper, we first describe these six BB emission datasets, the GEOS
model configuration and experimental designs, and observations in Sect. 2, then we show
comparisons of the biomass burning emission datasets and the resulting model simulated
AOD in Sect. 3. We discuss possible attributions of the differences between the six BB
emission datasets to the sources of uncertainty associated with the biomass burning
emissions and the aerosol modeling in Sect. 4. Conclusions and recommendations are
presented in Sect. 5.

## 2. Methodology
### 2.1 Six BB emission datasets
General information about each of the six biomass burning emission datasets investigated in this study, namely GFED3.1, GFED4s, FINN1.5, GFAS1.2, FEER1.0, and QFED2.4, is given below. Their main attributes, such as their spatial and temporal resolutions, the methods used to estimate burned area (where applicable), the method to derive emission coefficients (where applicable), and the references for the emission factors, are compared in Table 1. Overall, all datasets provide daily global biomass burning emissions since at least 2003.

### 2.1.1 *GFED3.1*
The total dry matter consumed by biomass burning in GFED3.1 (van der Werf et al., 2010) is estimated by the multiplication of the MODIS burned area product at 500-m spatial resolution (Giglio et al. 2010, for the MODIS era) and fuel consumption per unit burned area, the latter being the product of the fuel loads per unit area and combustion completeness. This estimation is conducted using the Carnegie–Ames–Stanford approach (CASA) biogeochemical modeling framework that provides estimates of biomass in various carbon "pools" including leaves, grasses, stems, coarse woody debris, and litter. Fuel loads in CASA are estimated according to carbon input information on vegetation productivity, and carbon outputs through heterotrophic respiration, herbivory, fires, and tree mortality (Giglio et al., 2010; van der Werf et al., 2010). Then, the biomass burning emission of a given species is calculated by multiplying the dry matter with an emission factor of that species (*EF*, with a unit of g species per kg dry matter burned). The *EF* used in GFED3.1 (and most of the other datasets) is mainly chosen from Andreae and Merlet (2001) and/or Akagi et al. (2011), but may also be obtained from various other sources. The GFED3.1 dataset can be accessed through the link: https://daac.ornl.gov/VEGETATION/guides/global_fire_emissions_v3.1.html.

### 2.1.2 *GFED4s*
Compared to GFED3.1, the latest GFED version, GFED4s, has a few significant upgrades as described in detail by van der Werf et al. (2017), including (1) additional burned area associated with small fires which were previously omitted by the burned area product but now are compensated by including the active fires to augment the burned area product MCD64A1 (Giglio et al., 2013; Randerson et al., 2012); (2) a revised fuel consumption parameterization optimized using field observations (e.g., van Leeuwen et al., 2014); (3) further dividing forest into temperate and boreal forest ecosystems and applying different sets of emission factors. Among the existing BB emission datasets, GFED4s has hitherto been the most widely used by modeling communities, such as the Coupled Model Intercomparison Project phase 6 (CMIP6, Van Marle et al., 2017) and AeroCom phase III experiment (https://wiki.met.no/aerocom/phase3-experiments). The link to the GFED4s dataset is http://www.globalfiredata.org.

### 2.1.3 *FINN1.5*
The FINN1.5 biomass burning emission dataset is developed from its previous version FINN1 (Wiedinmyer et al., 2011) with several updates. It uses satellite observation of active fire (with confidence level greater than 20%) and land cover from the MODIS

instruments onboard the NASA Terra and Aqua polar orbiting satellites, together with the
estimated fuel consumption to derive biomass burning emissions. The burned area in
each active fire pixel is assumed as 1 km$^2$, except for grasslands and savannas where it is
assigned a value of 0.75 km$^2$.  The fuel consumption at each fire pixel is estimated
according to its generic land use/land cover type (LULC) which is assigned using values
updated from Table 2 of Hoelzemann et al. (2004) in the various world regions based on
Global Wildland Fire Emission Model (GWEM). With the estimated burned area, fuel
consumption, and *EF* of individual species, the daily global open biomass burning
emissions of each species are then calculated at a 1 km spatial resolution. The FINN1.5
emissions dataset is archived at: http://bai.acom.ucar.edu/Data/fire/.
*2.1.4   GFAS1.2*
The GFAS (Kaiser et al., 2012) estimates dry matter combustion rate by multiplying FRP
and biome-specific conversion factors or emission coefficients $C_e$ (units: kg (dry matter)
per MJ). The global distribution of FRP observations are obtained from the MODIS
instruments onboard the Terra and Aqua satellites and then are assimilated into the GFAS
system. The gaps in FRP observations, which are mostly due to cloud cover and spurious
FRP observations of volcanoes, gas flares and other industrial activity, are corrected or
filtered in the GFAS system. Eight biome-specific $C_e$ are calculated by linear regressions
between the GFAS FRP and the dry matter combustion rate of GFED3.1 in each biome
(see Table 2 and Fig.3 in Kaiser et al., 2012). The biomass burning emission of a given
species is calculated by multiplying the dry matter with an emission factor of that species.
More information on the latest GFAS product can be found at
https://confluence.ecmwf.int/display/CKB/CAMS++Global+Fire+Assimilation+System+%28GF
AS%29+data+documentation.
*2.1.5   FEER1.0*
The FEER1.0 (Ichoku and Ellison, 2014) multiplies its emission coefficients $C_e$ with
MODIS FRP data that have been preprocessed and gridded in the GFAS1.2 analysis
system (Kaiser et al., 2012) to derive biomass burning aerosol emission rates. The $C_e$ in
FEER1.0 for smoke aerosol total particulate matter (TPM) was derived through zero-
intercept regression of the emission rate of smoke aerosol (i.e., $R_{sa}$) against the
corresponding FRP (Ichoku and Kaufman, 2005; Ichoku and Ellison, 2014) at pixel-level
within each grid. $C_e$ corresponds to the slope of the linear regression fitting. In the FEER
methodology, $R_{sa}$ is estimated through a spatio-temporal analysis of MODIS AOD data
along with wind fields from the NASA Modern-Era Retrospective Analysis for Research
and Applications (MERRA) reanalysis dataset (Rienecker et al., 2011). The smoke
aerosol $C_e$ in FEER1.0 is available at 1°×1° spatial resolution global grid, and covers
most of the land areas where fires have been detected by MODIS for at least 30 times
during the period 2003-2010 (Ichoku and Ellison, 2014) to ensure statistical
representativeness. In the current version of FEER1.0 emission dataset, $C_e$ for other
smoke constituents, say OC, at each grid cell are obtained by scaling the $C_e$ of smoke
aerosol according to the ratio of their emission factors, such as $EF_{oc}$ to $EF_{sa}$ (i.e., ratio of
emission factor for OC to that for total smoke aerosol). The FEER1.0 dataset is available
at http://feer.gsfc.nasa.gov/data/emissions/.

*2.1.6 QFED2.4*
In QFED (Darmenov and da Silva, 2015) biomass burning aerosol emissions are
estimated using gridded MODIS/Terra and MODIS/Aqua FRP and emission coefficients
$C_e$ which are the product of a constant value $C_0$ (1.37 kg per MJ, reported by Kaiser et al.,
2009), satellite factor and a biome-specific scaling factor. The scaling factors used in
QFED2.4, the version applied in this study, were obtained by comparing AOD from the
Goddard Earth Observing System model (GEOS) and MODIS aerosol product in
multiple sub-regions (Figure 4 in Darmenov and da Silva, 2015). These scaling factors
were further reduced to four values representative of fires in savanna, grassland, tropical
forests, and extratropical forests - 1.8, 1.8, 2.5, and 4.5, respectively. The QFED2.4 used
a sequential model with temporally damped emissions to estimate the emissions in cloudy
areas. QFED is the standard fire emissions dataset in the near real-time GEOS data
assimilation system and the MERRA-2 reanalysis (Randles, et al., 2017). QFED2.4
emissions are available from
https://portal.nccs.nasa.gov/datashare/iesa/aerosol/emissions/QFED/v2.4r6/.
**2.2 Application of the BB emission datasets in the NASA GEOS model**
**2.2.1 Description of the NASA GEOS model**
The GEOS model consists of an atmospheric general circulation model, a catchment-
based land surface model, and an ocean model, all coupled together using the Earth
System Modeling Framework (ESMF, Rienecker et al., 2011; Molod et al., 2015). Within
the GEOS model architecture, several interactively coupled atmospheric constituent
modules have been incorporated, including an aerosol and carbon monoxide (CO)
module based on the Goddard Chemistry Aerosol Radiation and Transport model
(GOCART, Chin et al., 2000, 2002, 2009, 2014; Colarco et al., 2010; Bian et al., 2010)
and a radiation module from the Goddard radiative transfer model (Chou and Suarez,
1999; Chou et al., 2001). The GOCART module used in this study includes
representations of dust, sea salt, sulfate, nitrate, and black and organic carbon aerosol
species. A conversion factor of 1.4 is used to scale organic carbon mass to organic
aerosol (OA), which is on the low end of current estimates (Simon and Bhave, 2012).
More discussion on this conversion factor can be referred to Sect. 4.3.
In this study the GEOS model (Heracles-5.2 version) was run globally on a cubed-sphere
horizontal grid (c180, ~50 km resolution) and with 72 vertical hybrid-sigma levels
extending from the surface to ~85 km for the year 2008. The reason we chose 2008 is
because it is the year assigned as a benchmark year by AeroCom community with which
this study is associated; it is also because the AeroCom Multi-model study of biomass
burning lead by Petrenko (mentioned in the introduction part) also chose 2008 as a focus
year. As such, the results from these two studies can be intercompared to draw some
synthesized conclusions. In addition, 2008 was chosen because it is a neutral ENSO year,
which represents normal burning conditions. The model was run in a "replay" mode,
where the winds, pressure, moisture, and temperature are constrained by the MERRA-2
reanalysis meteorological data (Gelaro et al., 2017), a configuration that allows a similar
simulation of real events as in a traditional off-line chemistry transport model (CTM) but
exercises the full model physics for, e.g., radiation, and moist physics processes. We used
the HTAP2 anthropogenic emissions (Janssens-Maenhout et al., 2015) that provides high-
spatial resolution monthly emissions. The BB emissions are uniformly distributed within
the boundary layer without considering the specific injection height of each plume. All
six BB emissions are daily emissions with the diurnal cycle prescribed in the model: the
maximum is around local noon, which is more prominent in the tropics, and is gradually
weakened in the extra-tropics (Randles et al., 2017). The natural aerosols are either
generated by the model itself (i.e., wind-blown dust and sea salt) or from prescribed
emission files (i.e., volcanic and biogenic aerosols).
**2.2.2 Experiment design**
In order to investigate the sensitivity of the modelled AOD to different BB emission
datasets, seven experiments were conducted with the GEOS model, differing only in the
source of biomass burning emissions. The first six runs are GFED3.1, GFED4s, FINN1.5,
GFAS1.2, FEER1.0, and QFED2.4, using the corresponding biomass burning datasets
described above in Section 2.1. A seventh run is called "NOBB," where the model is run
without including biomass burning emissions.
**2.4. AOD Observations**
**2.4.1 MISR retrievals**
We evaluated the simulated monthly AOD at 550 nm with the monthly level 3 total AOD
data at 558 nm wavelength from the Multiangle Imaging SpectroRadiometer sensor on
board the EOS-Terra satellite (Kalashnikova and Kahn, 2006; Kahn et al., 2010). We
used MISR version 23  data products (MISR v23, with filename tagged as F15_0032) in
half-degree, which can be downloaded from the website:
https://eosweb.larc.nasa.gov/project/misr/mil3mae_table.
**2.4.2 AERONET sites**
We also evaluated the modelled 3 hourly and monthly AOD at 550nm and Angström
Exponent (AE, 440–870 nm) with corresponding measurements from the ground-based
AErosol RObotic NETwork (AERONET, Holben et al., 1998) sites situated in biomass
burning source regions. AERONET Version 3 Level 2.0 data, which are cloud-screened
and quality-assured aerosol products with a 0.01 uncertainty (Giles et al., 2019), were
used in this study. The data can be downloaded from the website:
https://aeronet.gsfc.nasa.gov/new_web/download_all_v3_aod.html. The AERONET
AOD at 550nm is interpolated from the measurements at 440 and 675nm. AE is
calculated with AOD at 440 and 870nm. We compared model simulations with
AERONET data at 14 selected sites, representing the aerosol spatiotemporal
characteristics at the different biomass burning regions shown in Fig. 1. The 14 regions
were defined previously in GFED-related series of studies (e.g., Van der Werf et al.,
2006, 2010, and 2017). Some regions, such as Northern Hemisphere South America
(NHSA) and Equatorial Asia (EQAS), have no AERONET sites with data measured in
2008, thus we also used the average of multiple years or climatology of AERONET AOD
at each site for reference. Locations of these 14 selected AERONET sites are represented
by the numbered magenta dots in Fig.1.
**3. Results**

**3.1 Inter-comparison of the six biomass burning emission datasets**
The biomass burning OC emissions were compared throughout this study, since OC is the major constituent in fresh biomass burning smoke particles, with mass fractions ranging from 37% to 67% depending on fuel type (e.g., grassland/savanna, forests, or others), according to various studies based on thermal evolution techniques (Reid et al., 2005, part II, Table 2). These inter-comparisons were carried out in terms of both annual and seasonal variations in Sect. 3.1.1 and Sect. 3.1.2, respectively.

**3.1.1 Annual total**
Figure 2 shows the spatial distributions of annual total biomass burning OC emissions in 2008 from the six BB emission datasets. The regions with high emission of OC in Africa, boreal Asia, and South America are pronounced in all six BB emission datasets, albeit to different degrees. The regional differences of the annual total biomass burning OC emissions in different BB emission datasets can be appreciated more quantitatively in Fig. 3. Relevant statistics for the six BB emission datasets in the 14 regions are also listed at the top of the panel in Fig. 3, with the mean of the six BB emission datasets in the first row (*mean*). We also used three different measures to quantify the spread of the annual total from the six BB emission datasets: (1) standard deviation (*std*), (2) ratio of maximum to minimum (*max/min*), and (3) the coefficient of variation (*cv*, defined as the ratio of the *std* to the *mean*). The *cv* values for the 14 regions are also ranked in Fig. 3 for easy reference (e.g., a ranking of 1 means that this region shows the least spread among the six BB emission datasets, while a ranking of 14 indicates that this region has the largest spread).  The best agreements among the six emission datasets occurred in Northern Hemisphere Africa (NHAF), Equatorial Asia (EQAS), Southern Hemisphere Africa (SHAF), and Southern Hemisphere South America (SHSA), which have the top *cv* ranks (1-4) and relatively low *max/min* ratio (a factor of 3-4). The worst agreements occurred in the Middle East (MIDE), Temperate North America (TENA), Boreal North America (BONA), and Europe (EURO), which have the bottom *cv* ranks (14-11) and large *max/min* ratio (a factor of 66-10). This diversity was mostly driven by the QFED2.4 emission dataset, which estimated the largest emission amount for almost all regions (except EQAS), especially in MIDE where the BB emission from QFED2.4 is more than 50 times higher than those from the two GFED versions. Globally, the QFED2.4 dataset showed the highest OC emission of 51.93 Tg C in 2008, which was nearly four times that of GFED4s at 13.76 Tg C (the lowest among the six BB datasets).

Overall, two FRP-based BB emissions, QFED2.4 and FEER1.0, were a factor of 2-4 larger than the other BB datasets. This result is consistent with the findings of Zhang et al. (2014) over sub-Saharan Africa. It is worth noting that the BB emission amount of GFAS1.2 was close to that of GFED3.1, reflecting the fact that GFAS1.2 is tuned to GFED3.1(described in Sect. 2.1.4). Globally, FINN1.5 yielded more OC emissions than the two GFED datasets and GFAS1.2 (e.g., 40% larger than GFED4s). Regionally, FINN1.5 was generally comparable to the two GFED datasets in most of the regions, but was higher than them in the tropical regions, such as EQAS, Southeast Asia (SEAS), Central America (CEAM), and Northern Hemisphere South America (NHSA). Interestingly, FINN1.5 was even the largest among all six datasets over the EQAS region, which might be associated with its assumption of continuation of burning into the second

day in that region (to be discussed in Sect. 4.1.2). The global OC emissions from
GFED4s were lower than those from its GFED3.1 counterpart, although higher in several
other regions, such as TENA, CEAM, NHSA, Boreal Asia (BOAS) and Central Asia
(CEAS). Possible explanations for these differences among the six global BB emissions
datasets are provided in Sect. 4.1.
**3.1.2 Seasonal variation**
Biomass burning is generally characterized by distinct seasonal variations in each of the
14 regions and globally, as shown in Fig. 4. Overall, there were four peak fire seasons
across the regions: (1) During the boreal spring (March-April-May), fires peak in BOAS
mainly because of forest fires (see the contribution of different fire categories in Table 3
of van der Werf et al., 2017), in CEAM, NHSA, and SEAS because of savanna and
deforestation fires, and in Central Asia (CEAS) mainly due to the agricultural waste
burning to prepare the fields for spring crops. (2) During the boreal summer (June-July-
August), fires peak in BONA and TENA, mostly due to wildfires that occur under the
prevailing dry and hot weather, in EURO probably associated with the burning of
agricultural waste. In addition, we found that fire peaked in MIDE in the three FRP-based
datasets, i.e., QFED2.4, FEER1.0 and GFAS1.2. This might be associated with the failure
to filter out the gas flares from the FRP fire product, especially in QFED2.4 (Darmenov
and da Silva 2015). (3) During the austral spring (September-October-November), fires
peak in the southern hemispheric regions of SHSA, SHAF, and AUST, associated with
savanna burning (in addition to deforestation fires in SHSA). In SHSA, the two GFED
versions peaked in August, one month earlier than the rest; (4) During the boreal winter
(December and January), fires peak in NHAF, particularly along the sub-Sahel belt (Fig.
2), where savanna fires are associated with agricultural management and pastoral
practices across that region (e.g., Ichoku et al., 2016b). Overall, all six BB emission
datasets exhibited similar seasonal variations, although they differed in magnitude. In
particular, it is noteworthy that in EQAS, the annual OC emissions from GFED4s was
lower than that of GFED3.1 by 18%, but higher by a factor of two in the month of August
when peatland burning is predominant.
For reference, biomass burning black carbon (BC) emissions were also shown, but in the
supplement (Fig. S1, S2 for annual total and Fig. S3 for seasonal variation), which
exhibited similar features as OC. The amounts of biomass burning BC emission were
almost proportional to their OC counterparts (about 1/10 to 1/15 of OC).
**3.2 Comparison of model-simulated AOD with remote sensing data**
As in other similar situations where several different datasets are available to be chosen
from (e.g. Bian et al., 2007), a question that invariably comes to mind is: which BB
emission dataset is the most accurate or should be used in a given situation? In fact, it is
difficult to give a conclusive answer, as it is often challenging to measure the emission
rate of an active fire in real time or to disentangle the contribution of smoke aerosols
from the total atmospheric aerosol loading/concentration in observations. Therefore, in
this study we have implemented all six global BB emission datasets separately in the
GEOS model, and evaluated their respective simulated aerosol loadings. More
specifically, we compared the simulated AOD with the satellite-retrieved AOD data from
MISR (primarily to examine the spatial coverage) as well as with ground-based
measurements from AERONET sites near biomass burning source regions to examine the
seasonal variation. Our analysis was focused on the regional biomass burning peak
seasons, when smoke aerosol emissions dominate those from other sources, such as
pollution or dust. With such an effort to evaluate the sensitivity of the simulated AOD to
the different BB emission datasets, the results from this study may shed some light on
answering the aforementioned question, i.e., which BB dataset is the most accurate or
should be used in a given situation? We acknowledge that although the result from a
particular model (e.g., GEOS in this case) can potentially introduce additional uncertainty
through various complicated and non-linear procedures employed to calculate the AOD,
such as the modelled relative humidity and the related aerosol's hydroscopic growth
(Bian et al., 2009; Pan et al., 2015), still, evaluation of the model-simulated AOD has
proven to be a feasible approach to compare various BB emission datasets in reference to
the currently available observations (e.g., Petrenko et al. 2012; Zhang et al., 2014).
Aiming to evaluate the sensitivity of the modelled AOD to different BB emissions
datasets, we compared the spatial distribution of the GEOS model-simulated AOD with
MISR-retrieved AOD in Sect. 3.2.1 and with the AERONET measured AOD at 14
AERONET sites in Sect. 3.2.2. We also conducted an in-depth study at two AERONET
sites, Alta Floresta (in the southern hemisphere South America, SHSA) and Mongu (in
the southern hemisphere Africa, SHAF), as discussed in Sect. 3.3.
**3.2.1 Global spatial distribution**
Comparisons for September and April in 2018 are shown in Fig. 5 and Fig. 6
respectively, representing the peak biomass burning months in the southern hemisphere,
and many regions in the northern hemisphere, respectively. The MISR AOD is displayed
on the top left panel and the model biases (model minus MISR) from the seven individual
experiments are shown on the rest of the panels.
In September 2008, the high AOD observed from MISR (Fig. 5a) in the southern
hemisphere was mostly attributable to biomass burning. A large fraction of the southern
hemisphere Africa (SHAF) featured high AOD (greater than 0.5). The area-averaged
AOD over the entire SHAF was 0.331 (see Table S1 for the area-averaged MISR AOD in
each region). The observed AOD peaked in southern Congo (nearly 1.0) and gradually
decreased westwards. A large negative model bias (-0.283) was found in the NOBB run
over the region of SHAF (greenish shading in Fig. 5b; see Table S1 for the area-averaged
model biases in each region). The negative bias was reduced most significantly in the
QFED2.4 run (see Fig. 5h), to -0.044, followed by the FEER1.0 run (see Fig. 5g), to -
0.079, but only to a limited extent in GFED4s and GFAS1.2 (see Fig. 5d and f,
respectively), whose biases were still as large as -0.208.
In southern hemisphere South America (SHSA), where the area-averaged MISR AOD
was 0.188, the maximum AOD was ~0.7 in central Brazil (Fig. 5a). The negative bias
averaged over SHSA was -0.132 in the NOBB run (Fig. 5b). The bias was most
significantly reduced in the FEER1.0 run to -0.021 (Fig. 5g), but appeared overcorrected
in the QFED2.4 run to 0.020 (see reddish shading in Fig. 5h). The reduction of negative
bias was again the least in the GFED4s run (Fig. 5d) and the GFAS1.2 run (Fig. 5f),
whose biases were still as large as -0.081 and -0.080, respectively.
Being mixed with and often surpassed by other aerosol types, however, the contribution
of biomass burning aerosols to the total AOD is hardly distinguishable from those of
other sources in the peak biomass burning months in certain regions, such as April (Fig.
6) in the regions of Southeast Asia (SEAS), Central Asia (CEAS), and Boreal Asia
(BOAS). Such complicated situations lead to the difficulties in evaluating the BB
emission datasets with AOD observations, especially when the background AOD
represented by the NOBB run was already overestimated, for example, in the region of
CEAS, or when the MISR AOD was missing, for example, in the region of BOAS (where
MODIS AOD was missing as well, not shown).
**3.2.2 Seasonal variations of AOD at AERONET sites**
In order to better quantify the sensitivity of the simulated AOD to the six different BB
emission datasets, we further compared the simulated monthly AOD with the ground-
based AOD observations from AERONET stations by choosing one representative station
in each region (see Fig. 7: the panels representing the AERONET stations in Fig. 7 were
arranged in a way that their placements correspond to those of their respective regions in
Fig. 4 for easy reference). The exceptions are two regions NHSA and EQAS, where valid
AERONET observations could not be found during 2008. Thus, we used the multi-year
climatology of AOD at Medellin and Palangkaraya to represent NHSA and EQAS,
respectively. We also included the climatology of AERONET AOD in the other 12
AERONET sites for reference. As shown in Fig. 7, the annual cycle of AOD in 2008 at
available sites (brown thin bars) were similar to their respective climatology (light gray
thick bars) to within 0.05. The MISR AOD was plotted for reference as green diamond.
In this section, the modelled monthly mean AOD was calculated by averaging over the
modelled instantaneous AOD in each month; while the monthly AOD of AERONET and
MISR were simply calculated by averaging over available observations in each month.
Contributions from non-BB emissions to the total AOD are represented by NOBB
experiment (black line in Fig. 7). Runs with different BB emission datasets showed
almost identical AOD during non-biomass burning seasons at each selected AERONET
station in each region, thereby allowing their differences to be noticeable during the
biomass burning peak seasons. At Alta Floresta in Brazil (Fig. 7(5)), Mongu in Zambia
(Fig. 7(9)), and Chiang Mai Met Sta in Thailand (Fig. 7(12)), where the biomass burning
emissions dominated the peak AOD, almost all experiments underestimated AOD during
the respective peak biomass burning seasons. However, the fact is that the contribution of
non-BB AOD was usually more than that of BB AOD during the burning seasons at most
of the selected AERONET sites, except at the above three sites, Therefore, it is difficult
to disentangle the effect of biomass burning on the total AOD in most situations,
especially when the model has difficulty representing the non-BB AOD, leading, for
example, to overestimation at three high-latitude (> 55°N) AERONET sites (the three
panels in the top row of Fig. 7), i.e., Fort McMurray in Canada (Fig. 7(1)), Toravere in
Estonia (Fig. 7(6)), and Moscow_MSU_MO in Russia (Fig. 7(10)). However, it is
apparent that the simulated AOD based on QFED2.4 were overestimated during October
and November at Fort McMurray in USA, indicating that QFED2.4 overestimated BB
organic carbon emission during these two months. In general, at most of the AERONET
sites, the simulated AOD based on QFED2.4 were the highest and closest to AERONET
AOD during the corresponding peak of the biomass burning seasons, followed by
FEER1.0 and FINN1.5, and then GFED3.1, GFEDv4 and GFAS1.2.
**3.3 Case studies in biomass burning dominated regions**
In order to investigate the relationship between AOD and biomass burning emission in
the context of daily variation, we focused on two AERONET stations, namely, Alta
Floresta in Brazil and Mongu in Zambia during September, for the in-depth analysis in
this section. Biomass burning emissions are known to be dominant at these locations and
month, as estimated by Chin et al. (2009), who found that 50-90% of the AOD was
attributable to biomass burning emissions according to GOCART model simulations.
Based on other previous studies also, e.g., two studies with multiple BB datasets applied
to one model, Pereira et al. (2016) in southern hemisphere South America, and
Reddington et al. (2016) in tropical regions including southern hemisphere South
America as well as Africa, or the AeroCom Multi-model study (M. Petrenko, personal
communication) with one biomass burning emission dataset (i.e., GFED3.1) mentioned
earlier in the introduction, there appears to be a general consensus that the simulated
AOD is consistently underestimated over southern hemisphere South America and Africa
in many models. Therefore, in this study, we further examined these two sites: Alta
Floresta in Brazil and Mongu in Zambia. We calculated the 3-hourly AOD by sorting the
instantaneous AOD from both AERONET and model outputs for each day into eight
time-steps, namely, 0, 3, 6, 9, 12, 15, 18, and 21Z. The modelled monthly mean AOD
was calculated by averaging over the modelled 3-hourly AOD, which coincided with 3-
hourly AERONET AOD in that month. The detailed analyses are discussed below.
**3.3.1 Alta Floresta in Brazil (Southern Hemisphere South America, SHSA)**
The monthly mean AOD observed from AERONET at Alta Floresta is 0.47 during
September 2008 (Fig. 8a). It shows that the simulated AOD from all six experiments
captured the high aerosol episode observed in the AERONET dataset during September
13 (AERONET AOD is about 1.0-1.2). The simulation with QFED2.4 BB emission
produced the closest agreement with the AERONET observed AOD with an average
*ratio* of 1.00. In contrast, the simulated AOD with FEER1.0 (*ratio*=0.73), FINN1.5
(*ratio*=0.55), GFAS1.2 (*ratio*=0.42), GFED3.1(*ratio*=0.40), and GFED4s (*ratio*=0.36)
tended to be underestimated most of the time. All experiments showed relatively low skill
at capturing the temporal variability of the observed AOD at Alta Floresta (*corr*=0.24-
0.60). The Angström Exponent (AE: an indicator of particle size) from AERONET is
1.66 (not shown), indicating that small particles, most likely those from smoke,
dominated the total aerosol loading at Alta Floresta (Eck et al., 2001). All experiments
matched the observed AE (not shown).
The OC column mass loading (Fig. 8b) in each run resembled its corresponding AOD
(Fig. 8a), implying that the day-to-day variation of OC column mass loading in this dry
season dominates the simulated AOD in the model, rather than other factors such as
relative humidity (RH). The OC column mass loading is determined by the regional scale
of emission, transport, and removal processes of aerosols; the latter two processes being
the same across the six experiments, given that the same model configurations were used.
Therefore, the differences of OC column mass loading and thus AOD across the six
experiments are attributed to the different choices of biomass burning emission datasets.
Figure 8c shows the local biomass burning OC emissions (i.e., at the 0.5°×0.5°grid box
where this site is located) in the different biomass burning emission datasets. We found
that there was a large contrast in the local biomass burning OC emission between
September 25 (as high as 1-2 $\mu g\ m^{-2}\ s^{-1}$) and the other days (close to zero) across all six
experiments although in different degrees. Similar emission patterns are found when
averaged over nine or 25 surrounding grid boxes (not shown). Such sharp contrast was
completely absent in the simulated OC column mass density (Fig. 8b) and AOD (Fig. 8a).
All of the foregoing evidence collectively suggest that the temporal variations of AOD
(and aerosol mass loading) in Alta Floresta during the burning season do not directly
respond to the local BB emission at the daily and sub-daily time scales, but to the
regional emission. The regional emission is further adjusted by the multiple processes
determining the residence time of aerosols in air (typically a few days), such as the
regional scale transport and removal of aerosols. The MODIS-Terra true color image
overlaid with active fire detections (red dots) on September 13, 2008 (Fig. 8d) confirms
that there were no active fires (represented by red dots) detected at Alta Floresta (blue
circle), and thus the dense smoke over there during this peak aerosol episode must have
been transported from the upwind areas rather than from localized BB emission sources.
Therefore, accurate estimation of both the magnitude and spatial pattern of regional
emissions is quite important.
**3.3.2 Mongu in Zambia (Southern Hemisphere Africa, SHAF)**
The case of Mongu is different from that of Alta Floresta. Figure 9d reveals that there
were numerous active fire detections (represented by the red dots in this MODIS-Aqua
true color image) at and close to Mongu (blue circle) on September 12, 2008, one of peak
aerosol episodes as Figure 9a shows. The simulated AOD from all six experiments
captured two peak aerosol episodes observed from the AERONET dataset during
September 11-12 and September 2-3 (AOD about 1.0), albeit underestimated (Fig. 9a).
All experiments also underestimated the sustained aerosol episode after September 20.
However, all model experiments almost reproduced the AERONET AE value of 1.80
throughout September at this site (not shown), confirming that the dominance of the fine-
mode aerosol particles in smoke aerosols is captured by the model irrespective of the BB
emission dataset used.
The biomass burning OC emissions averaged over the grid box of Mongu exhibited
distinct daily variations in each BB dataset (Fig. 9c). Similar emission patterns are found
when averaged over nine or 25 surrounding grid boxes (not shown). At this site, the day-
to-day variations of AOD still cannot be totally explained by the corresponding local
emission at Mongu. For example, emission from FEER1.0 on September 17 is six times
higher than that on September 2 (Fig. 9c), but the simulated AOD on September 17 is
twice lower than that on September 2 (Fig. 9a). However, the magnitude of AOD at
Mongu in each experiment corresponded to the magnitude of BB emission at the regional
scale, since it is apparent that the overall higher regional BB emissions still resulted in
higher column mass loading and thus AOD. For instance, FEER1.0 and QFED2.4, which
have the largest monthly total biomass burning OC emission over the region of SHAF
among the six BB emission datasets during September (2.27 and 2.92 Tg mon$^{-1}$,
respectively, as shown in Fig. 4), corresponded to the highest AOD (*ratio*=49% and 46%,
respectively, as shown in Fig. 9a); while FINN1.5 and GFED4s, which represent the
lowest monthly mean biomass burning OC emission over the region of SHAF (0.87 and
0.85 Tg mon$^{-1}$, respectively, as shown in Fig.4), corresponded to very low AOD (15%
and 19% of the observed, respectively).
Although the temporal variation of the ambient RH may partially contribute to the day-
to-day changes of the emission-AOD relationship, the close resemblance between the
model simulated AOD and column OC mass loading (Fig. 9b) excludes such possibility.
These evidence therefore suggest that the temporal variations of AOD (and aerosol mass
loading) in Mongu, where daily local BB emissions were present instead, do not directly
respond to the local BB emission at the daily and sub-daily time scales during the burning
season either as the case in Alta Floresta. It further confirms the importance of accurate
estimation of both the magnitude and spatial pattern of regional emissions as mentioned
in the case of Alta Floresta. Therefore, over southern hemisphere Africa and southern
hemisphere South America, an enhancement of regional BB aerosol emission amounts in
all the BB emission datasets except for QFED2.4 in the latter region is suggested by this
study in order to reproduce the observed AOD level although to different degrees.
**4. Discussion**
The simulated AOD is biased low in biomass burning dominated regions and seasons
almost across all six BB emission datasets as demonstrated in this study. More
explanations on differences among the six BB emissions datasets are discussed in Sect.
4.1. Basically, the uncertainty of the simulated AOD could be attributable to two main
sources: (1) BB emissions-related biases; (2) Model-related biases. They are discussed in
Sections 4.2 and 4.3, respectively.
**4.1 The possible explanations of differences among the six BB emission datasets**
**4.1.1 Higher BB emissions estimated from QFED2.4 and FEER1.0**
This study has shown that the QFED2.4 and FEER1.0 BB emission datasets are
consistently higher than the others, with QFED2.4 being the highest overall. Some of the
possible reasons responsible for this difference include:
***Constraining with MODIS AOD***. The emission coefficients *(C$_e$)* used to derive biomass
burning aerosol emissions in both QFED2.4 and FEER1.0 are constrained by the MODIS
AOD, although in different ways (detailed in Sect. 2.1.6 and 2.1.5, respectively). This is
not the case for the other BB emission datasets. Although GFAS1.2 uses the same FRP
products as FEER1.0 in deriving dry mass combustion rate, its emission is tuned to that
of GFED3.1. QFED2.4 applied four biome-dependent scaling factors to the initial
constant value *C$_0$* when deriving its *C$_e$*, by minimizing the discrepancy between the AOD
simulated by the GEOS model and that from MODIS in corresponding biomes. The
resulting QFED2.4 scaling factors are 1.8 for savanna and grassland fires, 2.5 for tropical
forests, and 4.5 for extratropical forests (Darmenov and da Silva, 2015). This partially
explains its very high OC biomass burning emission over the extratropical regions of
TENA, BONA and BOAS relative to the other emission datasets (Fig. 2-4). However, the
high BB emission estimated by QFED2.4 is questionable during October and November
of 2008 in the region of BONA (Fig. 4) according to the evaluation of its resulting AOD
relative to the AERONET AOD at the Fort McMurray site (Fig. 7(1)). As for FEER1.0,
the process of deriving $C_e$ involved calculating the near-source smoke-aerosol column
mass with the MODIS AOD (total minus the background) for individual plumes, thereby
limiting influence from other emission sources (Ichoku and Ellison, 2014).
*__Fuel consumption.__* In general, the FRP-based estimation approaches, such as GFAS1.2,
QFED2.4, and FEER1.0, may enable more direct estimates of fuel consumption from
energy released from fires, without being affected by the uncertainties associated with the
estimates of fuel loads and combustion completeness (e.g., Kaufman et al., 1998;
Wooster et al., 2003, 2005; Ichoku and Kaufman, 2005; Ichoku et al., 2008; Jordan et al.,
2008). However, FRP from non-BB sources, such as the gas flare, could be mistakenly
identified as BB sources. One example is over bare land in the eastern border of Algeria
in MIDE (refer the land type to the website:
http://maps.elie.ucl.ac.be/CCI/viewer/index.php), where QFED2.4 shows high OC
emission contrary to expectation (see Fig. 2f). Thus, additional screening of FRP fire
product is required.
**4.1.2 Features of FINN1.5**
Globally, the FINN1.5 dataset is lower than QFED2.4 and FEER1.0, but larger than
GFAS1.2, GFED3.1 and GFED4s (Fig. 3). Although FINN1.5 can capture the location of
the large wildfires using the active fire products, the estimation of burned area is rather
simple without the complicated spatial and temporal variability in the amount of burned
area per active fire detection or variability in fuel consumption within biomes. For
example, it estimates 1 km$^2$ burned area per fire pixel for all biomass types except for
savanna and grassland where 0.75 km$^2$/fire pixel is estimated instead. That might partially
explain why FINN1.5 is extremely low in AUST, as suggested by Wiedinmyer et al.
(2011). Additionally, the FINN1.5 dataset is the least over boreal regions, such as in
regions of BOAS and BONA, where FINN1.5 is only 1/3 and 3/5 of GFED4s,
respectively. Large forest fires dominate in BOAS and BONA, such that the direct
mapping of burned area as done in GFED4s and GFED3.1 produces more biomass
burning emissions (van der Werf et al., 2017). On the other hand, the BB emission in
FINN1.5 dataset is relatively large near the equator. For instance, it is the largest among
the six datasets over the region EQAS, and the second largest over the regions of CEAM
and SEAS (see Fig. 3). This might be attributed to the smoothing of the fire detections in
these tropical regions to compensate for the limited daily coverage by the MODIS
instruments due to gaps between adjacent swaths and higher chances of cloud coverage in
tropical regions (Wiedinmyer et al., 2011). Thus, in FINN1.5, each fire detected in the
equatorial region only is counted for a 2-day period by assuming that fire continues into
the next day but at half of its original size.

### 4.1.3 Difference between GFED4s and GFED3.1

Globally and in some regions, biomass burning OC emission in GFED4s is lower than that in GFED3.1 (see Fig. 2-4), although the former has 11% higher global carbon emissions and includes small fires (van der Werf et al., 2017). There are a few possible reasons, of which two major ones are as follows: (1) For aerosols, the implementation of lower *EF* for certain biomes in GFED4s than in GFED3.1 reduces the aerosol biomass burning emissions. As for the savanna and grassland, for instance, the GFED4s dataset mainly applies *EF* value recommended by Akagi et al. (2011), which is 2.62 g OC per kg dry matter burned, 18% lower than the *EF* from Andreae and Merlet (2001) used in GFED3.1, which is 3.21 g OC per kg dry matter burned (see Table 2). The new estimation of *EF* is 3.0±1.5 g OC per kg dry matter burned as suggested by Andreae (2019). With it, the OC emissions in savanna and grassland can be slightly enhanced, but would still be lower than those in GFED3.1. (2) The improvement in including small fires in GFED4s over GFED3.1 is probably offset by the occasional optimization of fuel consumption using field observations for overall carbon emissions. For instance, the turnover rates of herbaceous leaf (e.g., savanna) are increased in GFED4s, leading to the lower fuel loading and thus lower consumption for this land-cover type in GFED4s (van Leeuwen et al., 2014; van der Werf et al., 2017). Therefore, the biomass burning OC emissions are lower in GFED4s over SHAF, NHAF, and AUST (Fig. 3 and 4), where ~88% of the BB carbon emission is from savanna and grassland (van der Werf et al., 2017).

On the other hand, there are regions in the northern hemisphere where GFED4s is higher than GFED3.1. For example, over CEAS and EURO, small fires associated with burning of agricultural residues contribute to 43.6% and 58.6% of the carbon emissions, respectively (van der Werf et al., 2017). In spite of the 30% reduction of the *EF* in these two regions, the effect of including small fires in GFED4s is greater, resulting in the biomass burning OC emission from GFED4s being twice as high as that from GFED3.1. Another example is in BOAS where the biomass burning OC emissions are 10% higher in GFED4s than in GFED3.1. This is likely attributable to the higher *EF* used in GFED4s for boreal forest fires than that in GFED3.1 (9.60 vs. 9.14 g OC per kg dry matter, see Table 2), where 86.5 % of the carbon emission is from the Siberian forest (van der Werf et al., 2017).

It is interesting that the yearly total biomass burning OC emission from GFED4s is 20% lower than that from GFED3.1 in EQAS (Fig. 4), even though the small fires are included and the *EF* of peatland and tropical forest are higher in the former (Table 2). By examining the monthly variations over EQAS (Fig. 4), however, we found that GFED4s is actually higher than GFED3.1 in August by a factor of two when peatland burning is predominant, but equal to or lower than GFED3.1 in other months, particularly in May, leading to the overall lower annual total value in GFED4s.

### 4.2 Sources of the uncertainty associated with biomass burning emissions

Uncertainty in any of the six BB emissions datasets considered in this study could have been introduced from a variety of measurement and/or analysis procedures, including: detection of fire or area burned, retrieval of FRP, emission factors (see Table 1), biome

types, burning stages, and fuel consumption estimates, some of which are explained in
detail below.
***Fire detection.*** Most of the current global estimations of biomass burning emissions are
heavily dependent on polar-orbiting satellite measurements from MODIS on Terra and
Aqua (e.g., MCD14DL, MOD14A1, MYD14A1, and MCD14ML as listed in Table 1).
The temporal and spatial resolutions of these measurements impose limitations on their
ability to detect and characterize the relevant attributes of fires, such as the locations and
timing of active fires and the extent of the burned areas. Each of the two MODIS sensors,
from which all of the major BB datasets derive their inputs, can only possibly observe a
given fire location twice in 24 hours, which leaves excessive sampling gaps in the diurnal
cycle of fire activity (Saide et al., 2015). Even for these few times that MODIS makes
observations at its nominal spatial resolution of 1 km at nadir, it has the potential to miss
a significant number of smaller fires (e.g. Hawbaker et al., 2008, Burling et al, 2011,
Yokelson et al., 2011), as well as to miss fires obstructed by clouds, and those located in
the gaps between MODIS swaths in the tropics (Hyer et al., 2009; Wang et al., 2018). In
addition, MODIS fire detection sensitivity is reduced at MODIS off nadir views, with
increasing view zenith angles especially toward the edge of scan, where its ground pixel
size is almost a factor of 10 larger that at nadir (Peterson and Wang, 2013; Roberts et al.,
2009; Wang et al., 2018), resulting in dramatic decreases in the total number of detected
fire pixels and total FRP (Ichoku et al., 2016b; Wang et al., 2018). Moreover, all
operational remote sensing fire products have difficulty accounting for understory fires or
fires with low thermal signal or peatland fires such as those in Indonesia, where
smoldering can last for months (Tansey et al, 2008). These issues can propagate into the
uncertainties of the emissions datasets that are dependent on active fire detection
products, especially those based on FRP, e.g., GFAS1.2 (Kaiser et al., 2012), FEER1.0
(Ichoku and Ellison, 2014), and QFED2.4 (Darmenov and da Silva, 2015). This issue also
affects FINN1.5 (Wiedinmyer et al., 2011), which derives the burned area by assuming
each active fire pixel to correspond to a burned area of 1 km$^2$ for most biome types (see
details in *Sect.* 2.1.3), and GFED4s, which uses burned area product for large fires but
derives burned areas for small fires using the MODIS active fire product.
On the other hand, although the sparse diurnal sampling frequency may not necessarily
be an issue for the MODIS burned area product, upon which some of the emission
datasets are based (*e.g.*, GFED3.1), burned area product may not account for small fires
due to its low spatial resolution of 500-m, which may limit the identification of small
burned scars such as those generated by small fires from crop lands. In addition, the
estimation of biomass burning emission based on the burned area product, e.g., GFED, is
subject to the uncertainty associated with the estimation of fuel load and combustion
completeness as mentioned earlier.
***Emission factor (EF).*** The *EF*, used for deriving individual particulate or gaseous
species of smoke emissions from burned dry matter in all major BB emission datasets,
heavily depends on the two papers by Andreae and Merlet (2001) and Akagi et al. (2011).
The authors of these two studies made significant contributions by compiling the values
of *EFs* from hundreds of papers. However, the EFs can have significant uncertainties
(Andreae, 2019), because each EF results from a particular experiment or field campaign.
Some *EF*s are derived from lab-based studies whereby samples of fuels are burned in
combustion chambers (e.g., Christian et al., 2003; Freeborn et al., 2008), where the
combustion characteristics can be very different from those of large-scale open biomass
burning and wildfires; and some *EFs* are derived from field campaigns, where the
measurement locations are often not close enough to the biomass burning source due to
personnel safety and other logistic factors (Aurell et al., 2019). As discussed earlier in
Sect.4.1.3, the discrepancy between GFED4s and GFED3.1 can be partially explained by
the fact that different emission factors were used to derive these two products (also see
Table 2). This situation will not change much even if the *EF* value from the latest
estimation by Andreae (2019) were used.
***Biome types.*** The uncertainty of estimating BB emission could be partially attributed to
different definitions of major biome types, because the scaling factor of emission
coefficient for the FRP-based BB datasets (i.e., GFAS and QFED), or the emission factor
used by all BB datasets will be assigned accord to the biome type where the fire event
occurs. The six BB emission datasets examined in this study have different definitions of
major biome types, for example, six major biome types applied in GFED4.1s (Table 1 in
van der Werf et al., 2017), but eight in GFAS1.2 (Table 2 in Kaiser et al., 2012), and only
four in QFED2.4 (Table 2 in Darmenov and da Silva, 2015). In particular, peat
combustion is an important emission source in some regions, such as Equatorial Asia
(e.g., Kiely et al., 2019). However, not all the emission datasets include peat biome or
consider its unique characteristics. Among the emission datasets based on burned areas,
GFED3.1 and GFED4s consider peat, but FINN1.5 does not. In QFED2.4 and FEER1.0,
the peat biome is not explicitly identified. Thus, such differences in the approaches to the
processing of contributions from specific biomes may contribute to the differences
between emission datasets in some regions.
***Burning stages.*** Most current BB emission datasets do not distinguish the different
burning stages, such as the flaming and smoldering stages that have distinctive emission
characteristics. Typically, flaming dominates the earlier stage of a fire while smoldering
dominates the later part. In the case of boreal forest fires, for example, about 40% of the
combustion originates from the flaming phase while 60% comes from the smoldering
phase (Reid et al., 2005). In addition, smoldering combustion produces more OC and CO
than flaming combustion; whereas flaming combustion produces more BC and carbon
dioxide ($CO_2$) than smoldering (e.g., Freeborn et al., 2008).
**4.3 Sources of the uncertainty associated with aerosol modeling**
The model-related biases in the GEOS model, which other models most probably also
suffer from, include, for example, inaccurate representations of horizontal and vertical
transport of aerosol by wind, fire emission plume height, and estimation of aerosol
removal in models. Furthermore, the production of secondary organic aerosol (SOA) in
biomass burning plumes, which has been observed in lab studies and ambient plumes
(e.g., Bian et al., 2017; Ahern et al., 2019), are missing in these GEOS simulations. Given
the sparsity of the measurements of surface and vertical concentrations at the global
scale, we implemented an approach to evaluate model simulation uncertainty globally
due to biomass burning aerosol emissions by evaluating the resulting AOD against those
from satellite data and AERONET measurements, following the studies by Petrenko et al.
(2012) and Zhang et al. (2014). We acknowledge the uncertainties in calculating AOD,
such as uncertainties associated with assumptions of aerosol size distribution, optical
properties, aerosol water uptake, and vertical distribution of aerosol (e.g., Reddington et
al., 2019). In addition, Ge et al. (2017) showed that the choice of different meteorological
fields can also lead to uncertainty in simulating the modelled aerosol loading. For
instance, meteorological fields from ECMWF and National Centers for Environmental
Prediction (NCEP) can yield a factor of two difference in the resulting surface $PM_{2.5}$
concentration during the fire season of September in the Maritime continents.
Furthermore, the ratio of OA to OC is 1.4 in our study, as first determined by White and
Roberts (1977). However, this OA/OC ratio of 1.4 is at the low end of the generally
suggested range of 1.2-2.5 (Turpin and Lim, 2001; Zhang et al., 2005; Bae et al., 2006;
El-Zanan et al., 2006; Aiken et al., 2008; Chan et al., 2010). Observations suggest that
OA/OC values of 1.6± 0.2 should be used for urban aerosols and 2.1± 0.2 for non-urban
aerosols (Turpin and Lim, 2001). Enhancing this ratio can obviously increase the
resulting AOD, but a more accurate measurement of this ratio during biomass burning is
needed.
**5. Conclusions and recommendations**
In this study, we compared six global biomass burning aerosol emission datasets in 2008,
i.e., GFED3.1, GFED4s, FINN1.5, and GFAS1.2, FEER1.0 and QFED2.4. We also
examined the sensitivity of the modelled AOD to the different BB emission datasets in
the NASA GEOS model globally and in 14 regions. The main results are summarized as
follows:
a. The biomass burning OC emissions derived from GFED3.1, GFED4s, FINN1.5,
GFAS1.2, FEER1.0, and QFED2.4 can differ by up to a factor of 3.8 on an annual
average, with values of 15.65, 13.76, 19.48, 18.22, 28.48, and 51.93 Tg C in 2008,
respectively. The biomass burning BC emissions can differ by up to a factor of 3.4 on
an annual average, with values of 1.76, 1.65, 1.83, 1.99, 3.66, and 5.54 Tg C in 2008,
respectively. In general, higher biomass burning OC and BC emissions are estimated
from QFED2.4 globally and regionally, followed by FEER1.0.
b. The best agreement among the six emission datasets occurred in Northern
Hemisphere Africa (NHAF), Equatorial Asia (EQAS), Southern Hemisphere Africa
(SHAF), and South Hemisphere South America (SHSA), where the biomass burning
emissions are predominant in determining aerosol loading, with the top coefficient of
variation rankings (1-4) and relatively low *max/min* ratio (a factor of 3-4). The least
agreement occurred in the Middle East (MIDE), Temperate North America (TENA),
Boreal North America (BONA), and Europe (EURO), with the bottom coefficient of
variation rankings (14-11) and large *max/min* ratios (a factor of 66-10. It seems that
the diversity among the six BB emission datasets is largely driven by QFED2.4,
which estimates the largest emission amount for almost all regions (except for
equatorial Asia).
c. In Southern Hemisphere Africa (SHAF) and Southern Hemisphere South America
(SHSA) during September 2008, where and when biomass burning aerosols are
dominant over other aerosol types, the amounts of biomass burning OC emissions
from QFED2.4 and FEER1.0 are at least double those from the remaining four BB
emission datasets. The AOD simulated by the NASA GEOS based on these two BB
emission datasets are the closest to those from MISR and AERONET, but still biased
low. In particular, at Alta Floresta in SHSA, they can account for 36%-100% of the
observed AOD, and at Mongu in SHAF, the AOD simulated with the six biomass
burning emission datasets only account for 15%-49% of the observed AOD. Overall,
during the biomass burning peak seasons at most of the representative AERONET
sites selected in each region, the AOD simulated with QFED2.4 is the highest and
closest to AERONET and MISR observations, followed by that of FEER1.0.
Considering that regional scale transport and removal processes as well as wind fields
are the same across the six BB emission experiments since they were run under the
same model configurations except for BB emission, it is evident that enhancement of
BB emission amounts in all six BB emission datasets will be needed (although to
different degrees) for the model AOD simulations to match observations, particularly
in SHAF (Mongu) and SHSA (Alta Floresta), except for QFED2.4 in SHSA.
Although the result of this study is partially model-dependent, nevertheless, it sheds
some light on our understanding of the uncertainty of the simulated AOD associated
with the choice of biomass burning aerosol emission datasets.
Based on the results of the current study, it is appropriate to make some
recommendations for future studies on improving BB emission estimation. Our
understanding of the complexity, variability, and interrelationships between different fire
characteristics (behavior, energetics, emissions) still need to be improved (Hyer et al,
2011). More accurate estimation of emission factors ($EF$) for different ecosystem types
and burning stages would greatly improve the emission overall, as demonstrated by the
discrepancy between GFED3.1 and GFED4s (see Sect. 4.1.3). The global BB emission
datasets driven by fire remote sensing and retrievals of FRP and burned-area products,
which have hitherto depended heavily on MODIS, can be augmented with products from
higher resolution sensors such as Visible Infrared Imaging Radiometer Suite (VIIRS),
and the global suite of geostationary meteorological satellites such as Meteosat (covering
Europe, Africa and the Indian Ocean), Geostationary Operational Environmental Satellite
(GOES, covering North, Central, and South America) and Himawari (covering east Asia,
southeast Asia, and Australia). Also, measurements from the recent field campaigns such
as WE-CAN (https://www.eol.ucar.edu/field_projects/we-can) and FIREX-AQ
(https://www.esrl.noaa.gov/csd/projects/firex-aq/science/motivation.html) are expected to
contribute toward advancing our knowledge of biomass burning emissions in North
America. The evaluation in this study has been solely based on remotely-sensed AOD
data, including retrievals from both satellite (MISR) and ground-based (AERONET)
sensors. Continuous mass concentration measurements are needed to validate the fire-
generated aerosol loading in specific contexts, such as in analyzing collocated surface
and vertical aerosol concentrations and composition, at least in the major BB regions.
**Data availability**
The GFED3.1 biomass burning dataset can be accessed through the link:
https://daac.ornl.gov/VEGETATION/guides/global_fire_emissions_v3.1.html. The link
to the GFED4s dataset is http://www.globalfiredata.org. The FINN1.5 emissions dataset
is archived at: http://bai.acom.ucar.edu/Data/fire/. The GFAS1.2 emissions dataset is
available at: https://apps.ecmwf.int/datasets/data/cams-gfas/. The FEER1.0 dataset is
available at http://feer.gsfc.nasa.gov/data/emissions/. The QFED2.4 can be downloaded
from the website :
https://portal.nccs.nasa.gov/datashare/iesa/aerosol/emissions/QFED/v2.4r6/. MISR level
3 AOD data can be downloaded from website:
https://eosweb.larc.nasa.gov/project/misr/mil3mae_table. AERONET Version 3 Level 2.0
data can be downloaded from the websites:
https://aeronet.gsfc.nasa.gov/new_web/download_all_v3_aod.html. The GEOS model
results can be obtained by contacting the corresponding author.

**Author contribution**
CI, MC, and XP conceived this project. XP conducted the data analysis and the model
experiments. XP and CI wrote the majority of this manuscript, and all other authors
participated in the writing process and interpretation of the results. HB, AD, PC and AS
helped with model set-up. CI, AD, and LE provided the biomass burning emission
datasets and interpretation of these datasets. TK, JW, and GC provided the help to apply
the biomass burning emission datasets in the model. CI and MC provided funding
supports.

**Acknowledgement**
MISR AOD data were obtained from the NASA Langley Research Center Atmospheric Science
Data Center. We thank the AERONET networks for making their data available. Site PIs and data
managers of those networks are gratefully acknowledged. We acknowledge the use of imagery
from the NASA Worldview application (https://worldview.earthdata.nasa.gov), part of the NASA
Earth Observing System Data and Information System (EOSDIS). Computing resources
supporting this work were provided by the NASA High-End Computing (HEC) Program through
the NASA Center for Climate Simulation (NCCS) at Goddard Space Flight Center. We also thank
the providers of biomass burning emission datasets of GFED, FINN, and GFAS. We
acknowledge supports from various NASA earth science programs, including the NASA
Atmospheric Composition Modeling, Analysis, and Prediction Program (ACMAP), the
Modeling, Analysis, and Prediction program (MAP), the Interdisciplinary Studies Program (IDS),
and Carbon Cycle Science program. CI is also grateful for partial support received during the
preparation of this article from the Educational Partnership Program of the National Oceanic and
Atmospheric Administration (NOAA), U.S. Department of Commerce, under Agreement No.
#NA16SEC4810006. TO is supported by NASA under Grant No. NNX14AM76G. We appreciate
the constructive comments from two reviewers, which help us to improve the quality of this
manuscript. XP also acknowledges the valuable suggestions from Dr. Dongchul Kim. The
contents of this article are solely the responsibility of the authors and do not represent the official
views of any agency or institution.

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

FIGURES

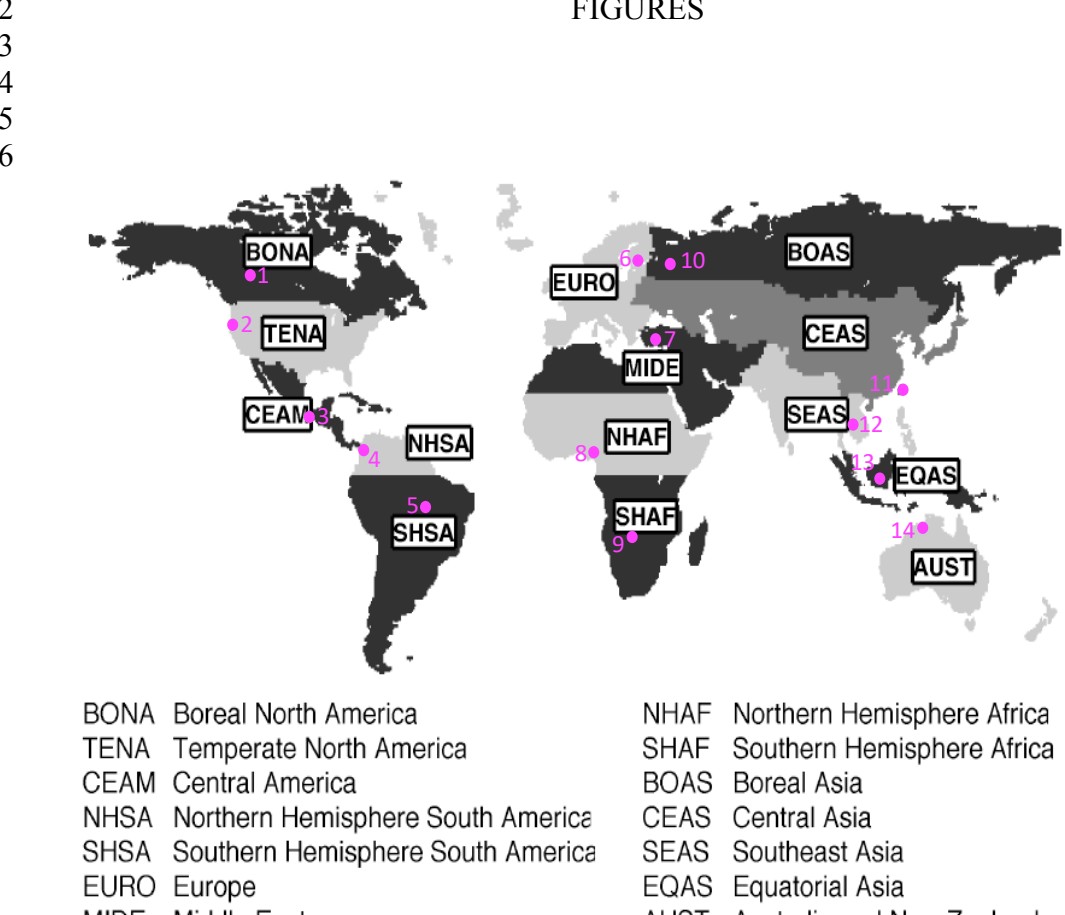

| | | | |
|---|---|---|---|
| BONA | Boreal North America | NHAF | Northern Hemisphere Africa |
| TENA | Temperate North America | SHAF | Southern Hemisphere Africa |
| CEAM | Central America | BOAS | Boreal Asia |
| NHSA | Northern Hemisphere South America | CEAS | Central Asia |
| SHSA | Southern Hemisphere South America | SEAS | Southeast Asia |
| EURO | Europe | EQAS | Equatorial Asia |
| MIDE | Middle East | AUST | Australia and New Zealand |

Figure 1. Map showing the 14 regions used in this study, following GFED regionalization defined by
Giglio et al. (2006) and van der Werf et al. (2006; 2017). The fourteen AERONET sites selected for
detailed analysis in the respective regions are represented by the numbered magenta dots. These
AERONET sites and the included data (years in parentheses) for calculating aerosol climatology are:
1-Fort McMurray (2005-2018), 2-Monterey (2002-2018), 3-Tuxtla Gutierrez (2005-2010), 4-
Medellin (2012-2016), 5-Alta Floresta (1993-2018), 6-Toravere (2002-2017), 7-IMS METU
ERDEMLI (1999-2017), 8-Ilorin (1998-2018), 9-Mongu (1997-2010), 10-Moscow MSU MO (2001-
2017), 11-EPA NCU (2004-2018), 12-Chiang Mai Met Sta (2007-2017), 13-Palangkaraya (2012-
2017), 14-Lake Argyle (2001-2017).

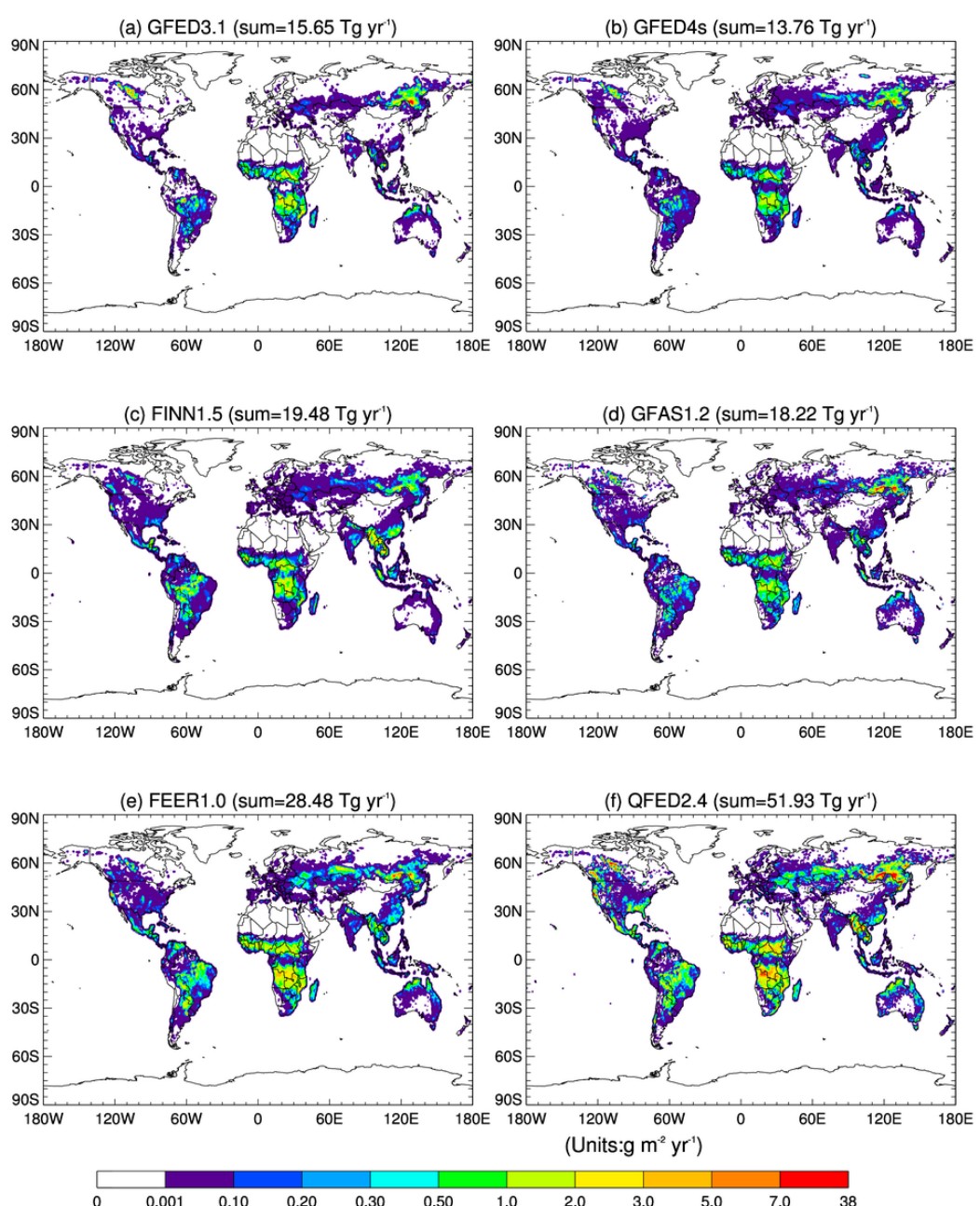

Figure 2. The spatial distribution of annual total organic carbon (OC) biomass burning emissions
for 2008 estimated by six biomass burning emission datasets (units: g m$^{-2}$ yr$^{-1}$). The global annual
total amount for each dataset in 2008 is indicated in the parentheses.

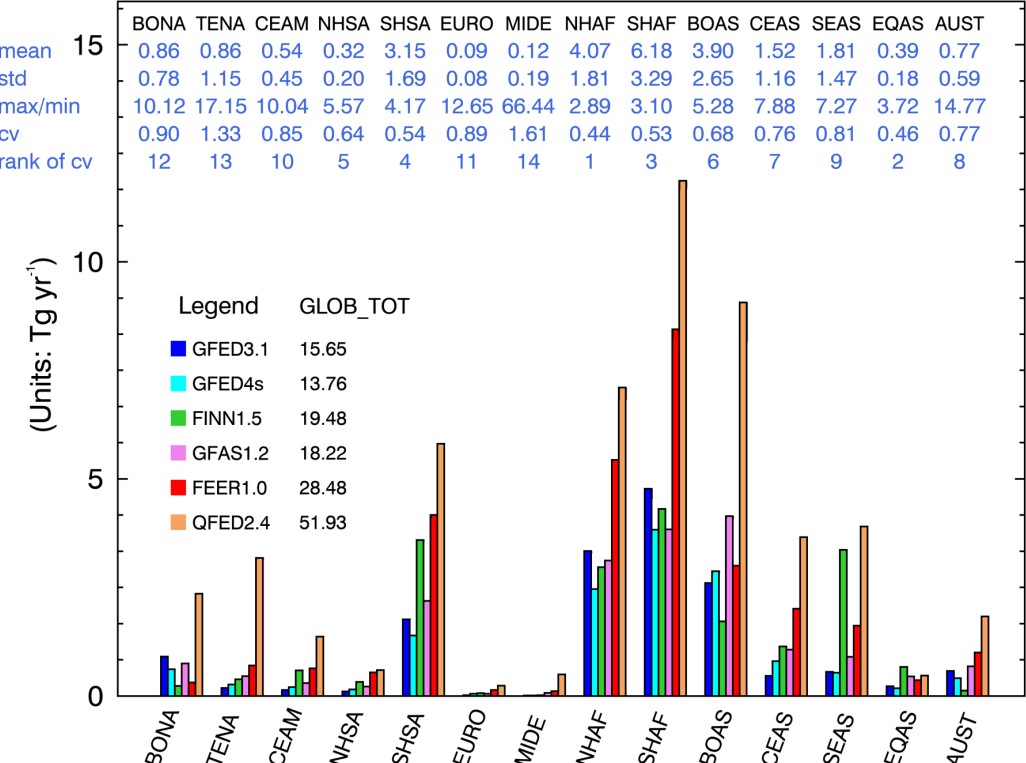

Figure 3. The regional annual total organic carbon (OC) biomass burning emissions for 2008 in
six biomass burning emission datasets in 14 regions (units: Tg yr$^{-1}$). The global annual total
amount is listed after the name of each dataset (GLOB_TOT). Relevant statistics for the six BB
emission datasets in each region are also listed at the top of the panel in blue under the short name
of each region, with the mean of the six BB emission datasets in the first row. Three different
methods to measure the spread of the six BB emission datasets are shown as well: one absolute
method, i.e., the standard deviation (std) in the second row, and two relative methods, i.e., the
ratio of max to min (i.e., maximum/minimum) shown in the third row, and the coefficient of
variation (cv), defined as the ratio of the std to the mean, in the fourth row. The rankings of the
regions reflecting the spread of the BB emissions datasets according to cv are shown in the fifth
row (i.e., a ranking of 1 means that this region shows the least spread among the six BB
emissions datasets, while a ranking of 14 indicates that this region has the largest spread among
the 14 regions).

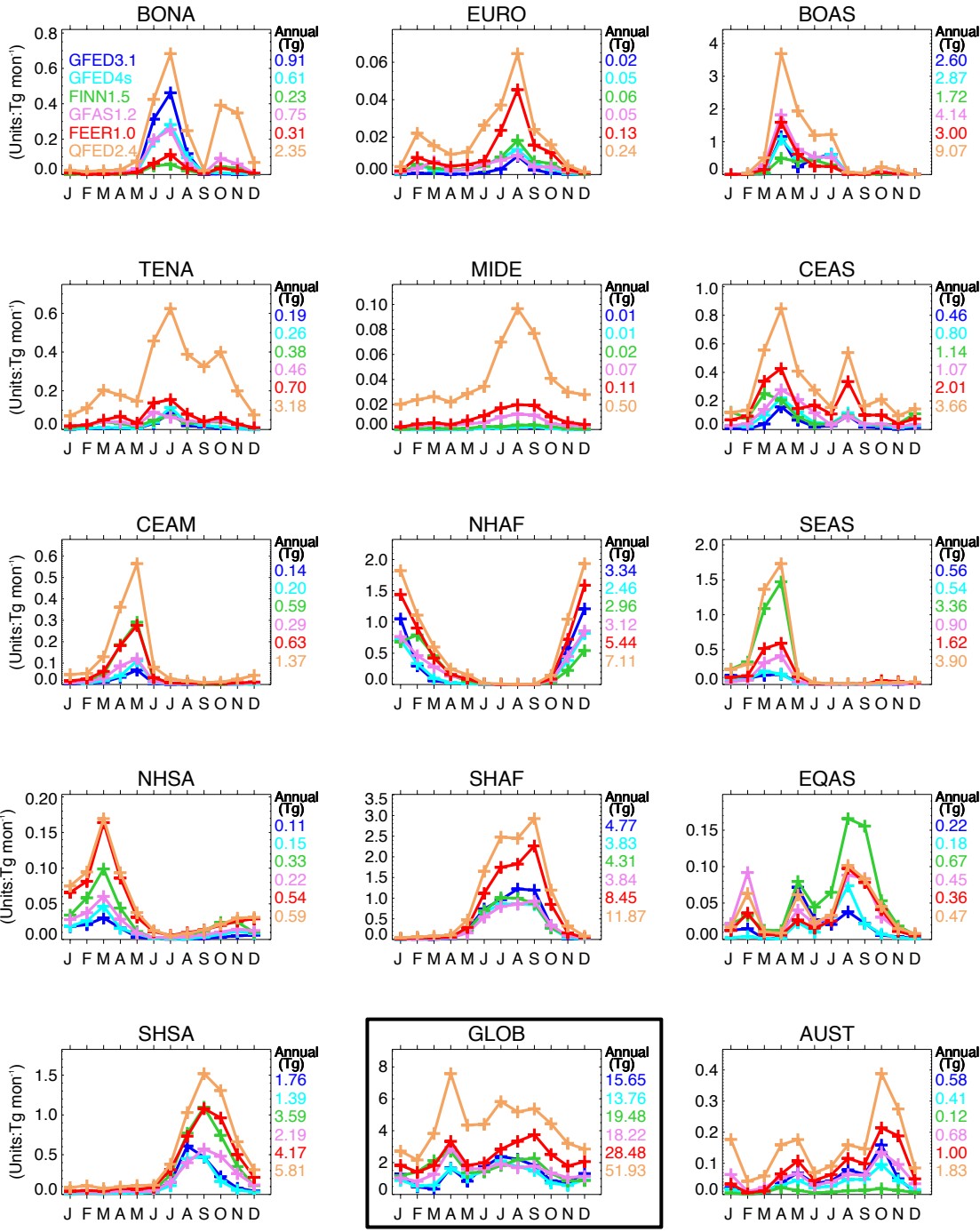

Figure 4. Monthly variation of organic carbon (OC) biomass burning emissions for 2008 in six
biomass burning emission datasets in 14 regions and the globally (i.e., GLOB, highlighted with a
black box).  The annual total emission is listed on the right side of each panel.


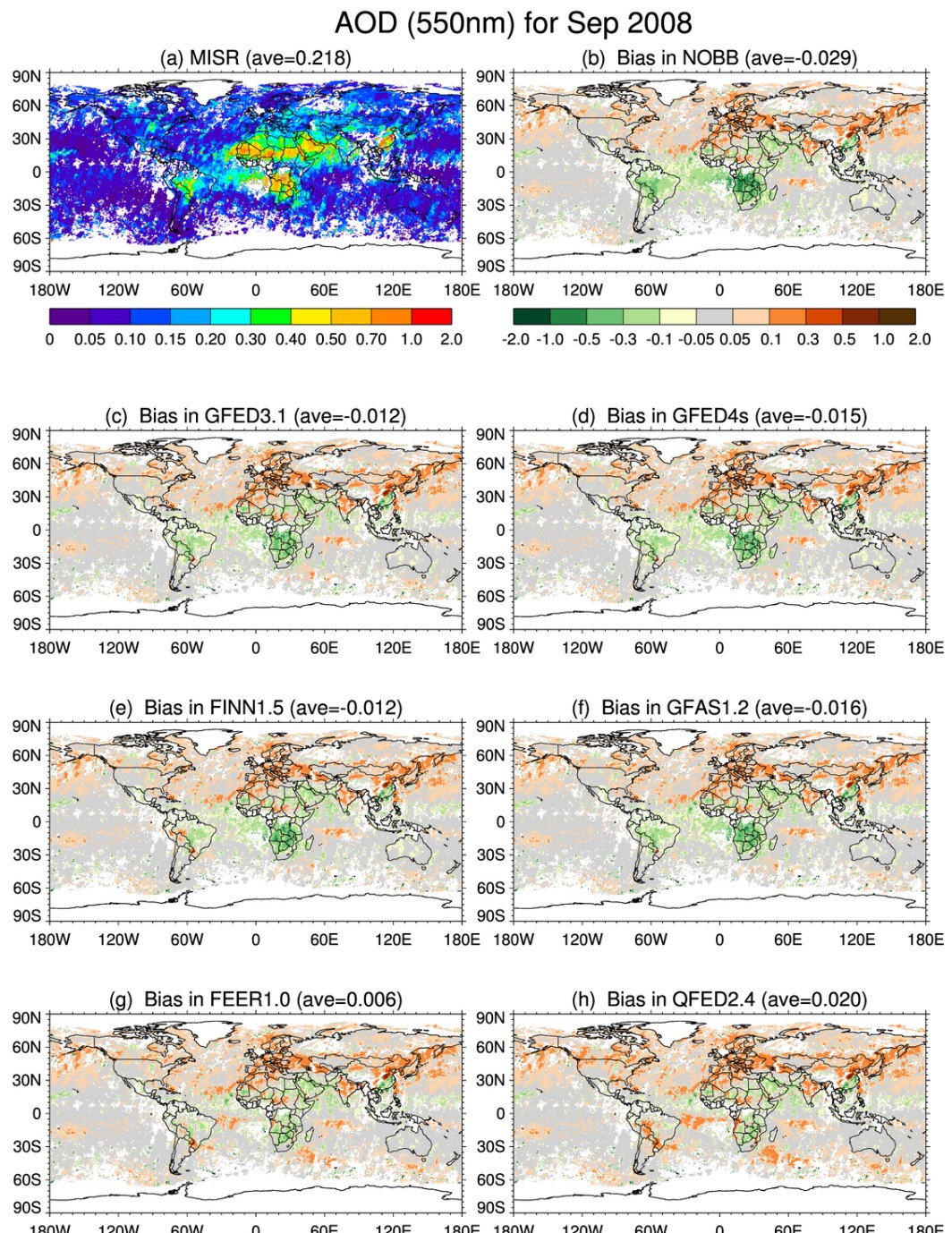

Figure 5. (a) The spatial distribution of monthly mean AOD at 558nm for September 2008 from
MISR with the white color representing missing value. The global average value (ave) is shown
in parentheses. (b)-(h) are for GEOS model biases (i.e., model at 550nm minus MISR at 558nm)
in seven model experiments, i.e., bias in (b) NOBB, (c) GFED3.1, (d) GFED4s, (e) FINN1.5, (f)
GFAS1.2, (g) FEER1.0, (h) QFED2.4.

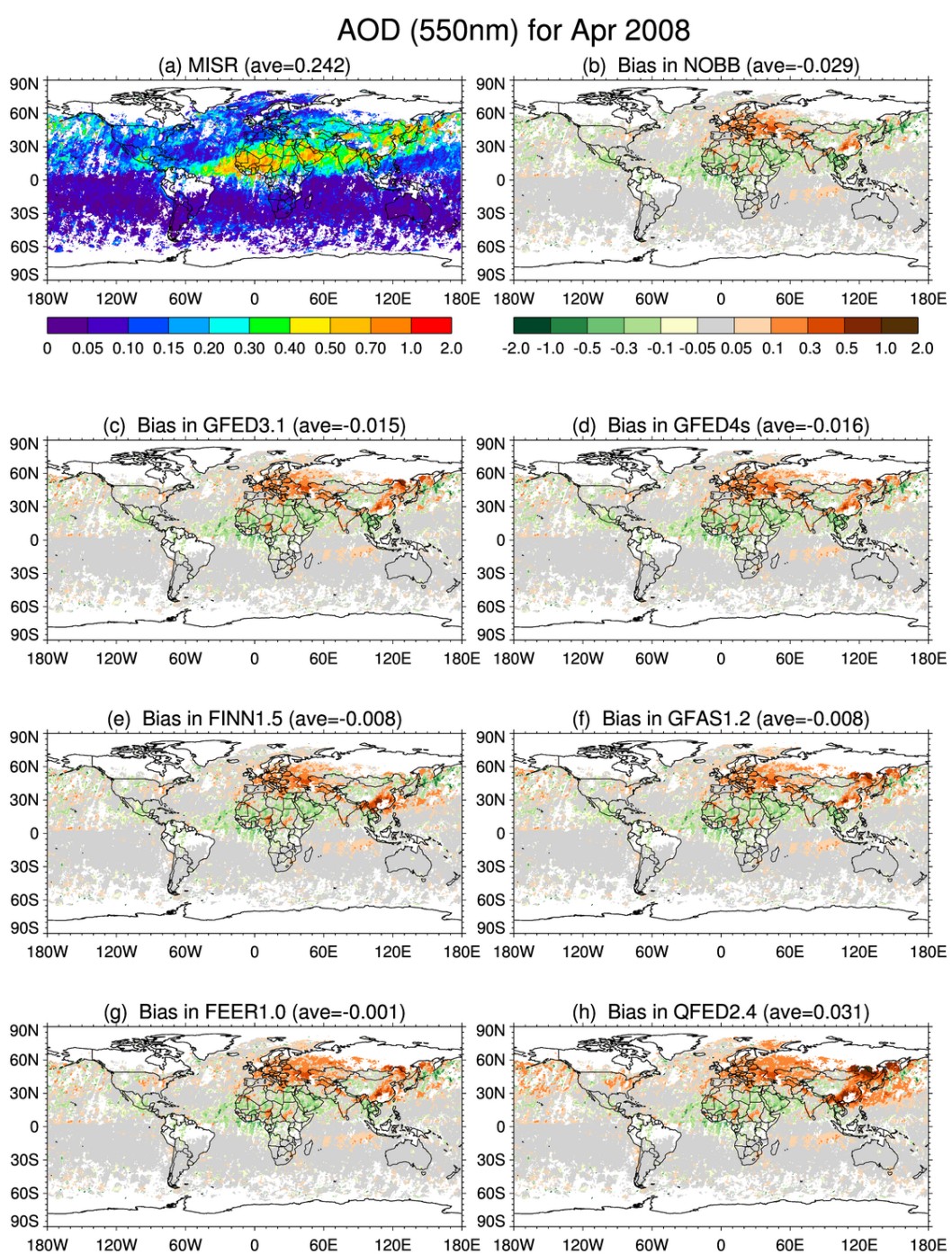

AOD (550nm) for Apr 2008

(a) MISR (ave=0.242)
(b) Bias in NOBB (ave=-0.029)
(c) Bias in GFED3.1 (ave=-0.015)
(d) Bias in GFED4s (ave=-0.016)
(e) Bias in FINN1.5 (ave=-0.008)
(f) Bias in GFAS1.2 (ave=-0.008)
(g) Bias in FEER1.0 (ave=-0.001)
(h) Bias in QFED2.4 (ave=0.031)

Figure 6. Same as Figure 5 except for April 2008.

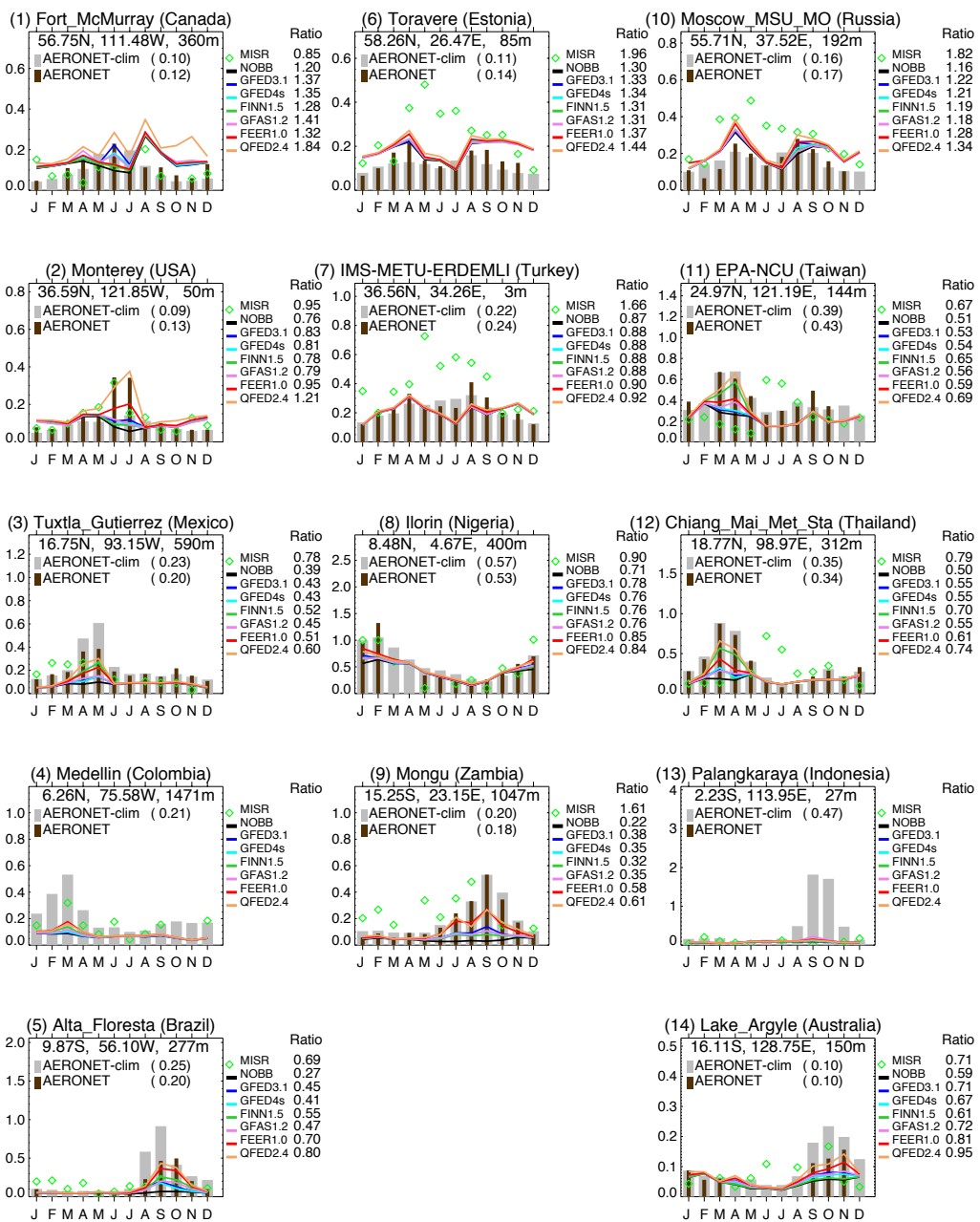

**Monthly AOD (550nm) at AERONET sites for 2008**

Figure 7. Monthly variation of AOD (at 550nm wavelength) for 2008 over 14 AERONET sites
selected from the respective 14 regions (with the country of each site indicated in parentheses).
The climatology of AERONET AOD (i.e., AERONET-clim) is represented by light gray thick
bars and the monthly mean AERONET AOD for 2008 by brown thin bars, with their
corresponding annual mean values shown in parentheses. MISR monthly mean AOD at 558nm is
represented by the green diamonds, and the seven GEOS experiments with different biomass
burning emission options are represented by the lines in different colors. The corresponding
annual ratios (Ratio=model/AERONET) listed on the right-hand side of each panel are estimated
by averaging over monthly ratios.


*Alta Floresta (September 2008)*

(a) AOD

(b) OC column mass density

(c) OC biomass burning emission

(d) MODIS Terra 2008-09-13

Figure 8. Characteristics of the observed and the simulated aerosols at Alta Floresta during
September 2008: (a) The 3-hourly time series of AOD at 550nm. The AERONET AOD is
represented by vertical gray bars, and the outputs from the six model experiments are represented
by the color curves. The relevant statistics are listed: *ave* is the monthly average, *ratio* is the
fraction of the simulated to the observed AOD at all observed hours, *corr* is correlation
coefficient between the observed and the simulated AOD, and *rmse* is root mean square error. (b)
The 3-hourly time series of OC column mass density over the grid box where Alta Floresta is
located (units: 1.e-06 kg m$^{-2}$ or mg m$^{-2}$). (c) Same as (b) but for biomass burning OC emission
rate (units: 1.e-09 kg m$^{-2}$ s$^{-1}$ or µg m$^{-2}$ s$^{-1}$). (d) MODIS-Terra true color image around Alta
Floresta on September 13, 2008, overlaid with the active fire detections in red dots (Image credit:
https://aeronet.gsfc.nasa.gov/cgi-bin/bamgomas_interactive and
https://worldview.earthdata.nasa.gov).

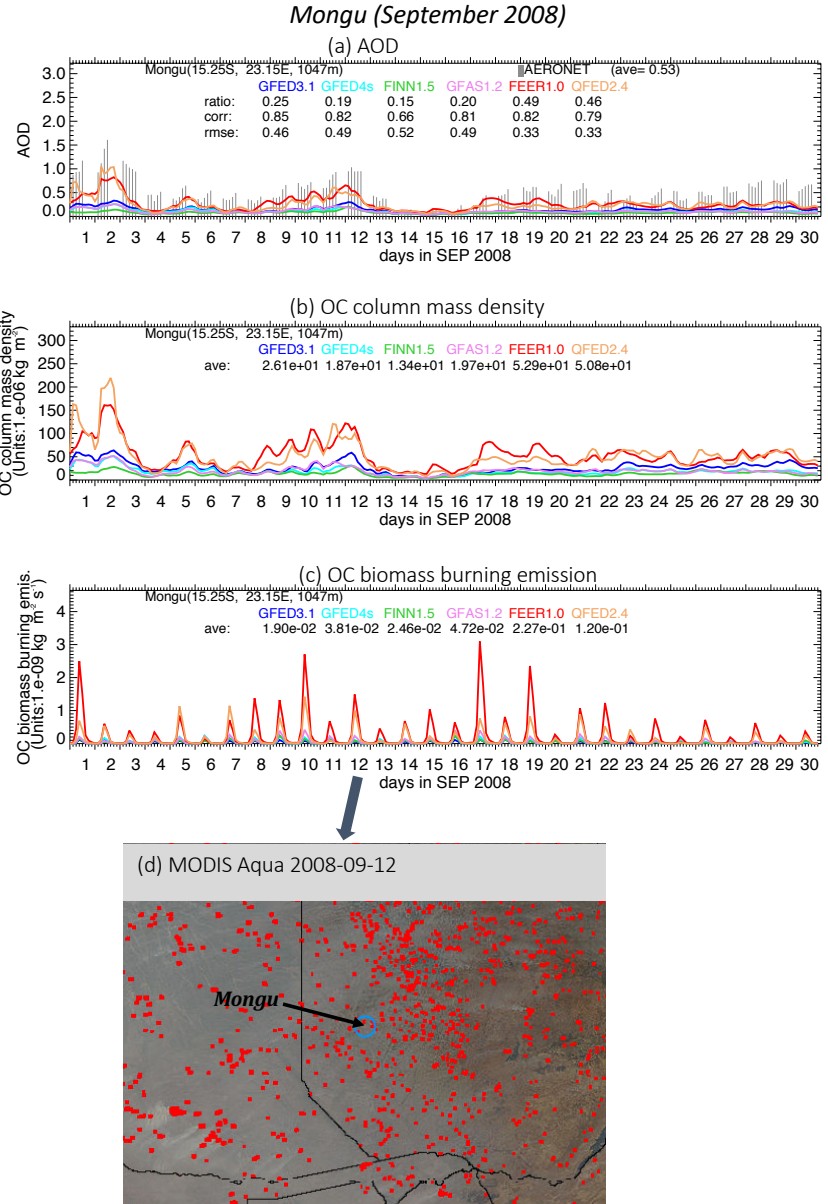

Figure 9. Characteristics of the observed and the simulated aerosols at Mongu during September
2008: (a) The 3-hourly time series of AOD at 550nm. The AERONET AOD is represented by
vertical gray bars, and the outputs from the six model experiments are represented by the color
curves. The relevant statistics are listed: *ave* is the monthly average, *ratio* is the fraction of the
simulated to the observed AOD at all observed hours, *corr* is correlation coefficient between the
observed and the simulated AOD, and *rmse* is root mean square error. (b) The 3-hourly time
series of OC column mass density over the grid box where Mongu is located (units: 1.e-06 kg m$^{-2}$
or mg m$^{-2}$). (c) Same as (b) but for biomass burning OC emission rate (units: 1.e-09 kg m$^{-2}$ s$^{-1}$ or
µg m$^{-2}$ s$^{-1}$). (d) MODIS-Aqua true color image around Mongu on September 12, 2008, overlaid
with the active fire detections in red dots (Image credit: https://aeronet.gsfc.nasa.gov/cgi-
bin/bamgomas_interactive and https://worldview.earthdata.nasa.gov).

**Table 1.** Summary of six biomass burning emission datasets during MODIS-era (i.e., 2000-present)

| a. Burned-area based approaches | | | | | |
|---|---|---|---|---|---|
| **BB Emission Dataset** | Original Grid | Time-Frame/ Frequency | Burned Area | Active Fire Product | Fuel Consumption | Emission Factor |
| **GFED3.1** | 0.5°×0.5° (lon×lat) | 2000-2012/ 3-hourly, daily, monthly | MOD09GHK and/or MYD09GHK | Gridded composite L3 fire product MOD14A1 and/or MYD14A1 | Estimated in CASA by product of fuel load and combustion completeness | Mainly from Andreae and Merlet (2001) with annual updates |
| **GFED4s** | 0.25°×0.25° (lon×lat) | 2000-2016/ 3-hourly, daily, monthly | Daily MCD64A1 product in Collection 5.1 at 500m spatial resolution | L3 MOD14A1 and MYD14A1; fire location product MCD14ML | Revised CASA by optimizing parameterization, reorientation of fuel consumption in frequently burned landscapes | Mainly from Akagi et al. (2011), supplemented by Andreae and Merlet (2001) and other |
| **FINN1.5** | 1km² | 2002-2015/ daily | Estimated by active fire counts: 0.75 km² for savannas at each fire pixel, 1km² for other types | MODIS NRT active fire product (MCD14DL) | Assigned according to the global wildland fire emission model (Hoelzemann et al., 2004) with updates | Mainly from Andreae and Merlet (2001) and Akagi et al. (2011), with updates through 2015 |

| b. FRP based approaches | | | | | |
|---|---|---|---|---|---|
| **BB Emission Dataset** | Original Grid | Time-Frame/ Frequency | FRP | Emission Coefficient ($C_e$) | Emission Factor |
| **GFAS1.2** | 0.1×0.1 (lon×lat) | 2003-Present/daily | Assimilation of level 2 MOD14 and MYD14 FRP | Linear regression between FRP dry matter and combustion rate of GFED v3.1 in eight biome types. | Mainly from Andreae and Merlet (2001) with updates from literatures through 2009 |
| **FEER1.0** | 0.1×0.1 (lon×lat) | 2003-Present/ daily, monthly | From GFASv1.2 (Kaiser et al., 2012, see above) | Linear regression between FRP and total particulate matter (PM) emission rate estimated from MODIS AOD at each grid. | Andreae and Merlet (2001) with updates provided by Andreae in 2014 |
| **QFED2.4** | 0.1×0.1 (lon×lat) | 2000-Present/ daily, monthly | Level 2 fire products MOD14/MYD14 | Based on comparison of MODIS and GEOS AODs and obtaining 4 biome-specific emission coefficients. | Andreae and Merlet (2001) |

CASA: Carnegie-Ames-Stanford-Approach biogeochemical

**Table 2.** Comparison of emission factor (units: g species per kg dry matter burned) used by GFED3.1[1] and GFED4s [2] (listed in the upper and lower part of the
cell respectively, bold if GFED4s is larger).

| | Savanna and Grassland | Tropical Forest | Temperate Forest [3] | Boreal forest [3] | Peat Fires [4] | Agricultural Residues |
|---|---|---|---|---|---|---|
| **OC** | 3.21<br>2.62 | 4.30<br>**4.71** | 9.14<br>**9.60** | 9.14<br>**9.60** | 4.30<br>**6.02** | 3.71<br>2.30 |
| **BC** | 0.46<br>0.37 | 0.57<br>0.52 | 0.56<br>0.50 | 0.56<br>0.50 | 0.57<br>0.04 | 0.48<br>**0.75** |
| **SO₂** | 0.37<br>**0.48** | 0.71<br>**0.40** | 1.00<br>**1.10** | 1.00<br>**1.10** | 0.71<br>0.40 | 0.40<br>0.40 |
| **PM₂.₅** | 4.94<br>**7.20** | 9.05<br>**9.10** | 12.84<br>**12.90** | 12.84<br>**15.30** | 9.05<br>**9.10** | 8.25<br>6.30 |
| **CO₂** | 1646<br>**1686** | 1626<br>**1643** | 1572<br>**1647** | 1572<br>1489 | 1703<br>1703 | 1452<br>**1585** |
| **CO** | 61<br>**63** | 101<br>93 | 106<br>88 | 106<br>**127** | 210<br>210 | 94<br>**102** |

[1] Mainly from Andreae and Merlet (2001) with annual updates
[2] Mainly from Akagi et al. (2011), supplemented by Andreae and Merlet (2001) and other sources
[3] GFED4s (van der Werf et al., 2017) further divides extra-tropical forest in GFED3 (van der Werf et al., 2010) into temperate forest and boreal forest.
[4] Based on Christian et al. (2003) for $CO_2$ and CO.