# Peer review of "Six Global Biomass Burning Emission Datasets: Inter-comparison and Application in one Global Aerosol Model"

_Atmospheric Chemistry and Physics, 2019_

## Referee Comment (RC1) · Anonymous Referee #1 · 16 Aug 2019

The authors have run the same atmospheric model with 6 different biomass burning (BB) inventories and analysed the differences using AOD and aeronet. These differences are often substantial and to some degree the authors have pointed to reasons why those differences exist. I feel the paper helps other modellers in understanding where some of the uncertainties in biomass burning emissions originate from but at the same time I feel the reader is left a bit wondering what the main messages are in the end. Ideally one would come up with recommendations about when and where to use a certain dataset, or when and where to avoid those. But given that the dataset to evaluate the results is also used to construct some this may be too much asked. Please find below a number of suggestions to further improve the paper

[Figure]

First sentence in introduction is spelled a bit awkward, please break up in two. Likewise for the second paragraph (L79).

159: Not sure why that small fire paper is cited in the GFED3 description

208: Kaiser et al. . ., -> Kaiser et al.,

The link on L213 does not work, at least not on my two computers

L282: I am a bit surprised that BB aerosols are injected near the surface. There is quite a bit of literature showing the importance of injection heights in for example the Boreal region

L297: So basically you use the same AOD data that was used to construct one of the BB inventories to evaluate a suite of models. That just doesn't feel right and requires careful explanation why this is done and what the consequences are

L305: I feel this is more useful and scientific sound; evaluate the various inventories with independent data

L393: But isn't April outside the main fire season in EQAS? In other words, if emissions are very low then a factor two difference (for example due to the detection of small fires in GFED4) is not that noteworthy I guess

L402: This is indeed a key question and I doubt we will make much progress as long as we keep using one single dataset to constrain emissions. Broadly speaking, the "gas community" (CO, NO2) has shown that the traditional inventories do reasonably well while the "aerosol community" has shown for over 10 years now that the emissions of those inventories are too low to reconstruct measured AOD. It would be very nice if someone would address why those two communities come to different conclusions.

L416 lights -> light

L419: GOES -> GEOS

[Figure]

L452: This is a bit confusing, I don't think emissions peaked in April but you found elevated AOD levels due to burning

L467: Given the very large interannual variability, especially in EQAS, this should be avoided. Please scale with active fire detections or so

L529: Now shown -> Not shown (I guess)

L624: This could be a place where this paper could make a difference. Given that the emission factors used in the various datasets are not wildly different, the variability stems from variability in dry matter fuel consumption. GFED has been tuned to match measured fuel consumption, how about FEER and QFAS? Are their levels of fuel consumption (per unit burned area that is) similar to literature-based values? I understand that the FRP approach aims to avoid burned area but these datasets are becoming better constrained and by dividing fuel consumption from FEER and QFED with burned area there could be a useful constraint. Right now we compare AOD with AOD-derived datasets and that just does not help us further I am afraid

L731: Actually most of the emission factors are from actual fires, not from lab-based measurements.

---

## Referee Comment (RC2) · Mark Parrington (Referee) · 17 Sep 2019

The manuscript presents a comparison of biomass burning emissions estimated using satellite observations of active fires including burnt area and fire radiative power. Evaluation of the different emissions datasets is performed by application in a global aerosol model and comparing the relative changes in the organic matter aerosol fields over MODIS satellite and AERONET ground-based observations of aerosol optical depth (AOD). The authors acknowledge the limitations of the nature of a model-specific study like this but the inter-comparison is very thorough and provides valuable, and timely, insights into variability of estimating biomass burning emissions for application in mod-

els. The manuscript is well written and in the scope of Atmospheric Chemistry and Physics, and I recommend it for publication subject to the authors addressing the comments below.

General comments:

Discussion of uncertainties in emission factors – would the known underestimate of PM emission factors, especially for peat fires in South East Asia, impact on the model AOD? https://www.mdpi.com/2072-4292/10/4/495/htm or https://agupubs.onlinelibrary.wiley.com/doi/pdf/10.1029/2017JD027827

Specific comments/questions:

Page 4, line 107: specify the multi-model study (is it "The AeroCom multi-model study"?).

Page 9, lines 365-367: could it be the case that the two day persistence in FINN1.5 is more representative of peat fires which may be more prevalent in EQAS?

Page 9, section 3.1.2: it may be useful to describe briefly why 2008 was chosen to investigate the seasonal variation. Does each emissions dataset capture inter-annual variability in the same way?

Page 10, line 409: "with each BB emission dataset instead" is repeating the earlier part of the sentence.

Page 11, line 433: change "peaking" to "peak".

Page 11, section 3.2.1: it may be useful to a reader to give the names of each region as well as the acronym.

Page 12, section 3.2.2: it may be useful to give the country of the named AERONET sites, which is more intuitive to understanding the geography than giving just the regions.

Page 12, lines 487-488: "in each respective region".

Page 12, line 488: change "At most other AERONET" to "At most of the other AERONET".

Page 13, line 534: "resembled with" should be "resembled".

Page 13, lines 537-538: "All of these evidences" should be "All of this evidence".

Page 13, line 539: should "respond" be "correspond".

Page 13, line 543 (and other locations): would using "active fire detections" rather than "fire hotspots" be a more scientific way of describing this?

Page 13, line 553: "over entire" should be "over the entire".

Page 14, line 566: should "emitted from smoke aerosols" be "emitted as smoke aerosols"?

Page 14, line 567: change "These evidences" to "This".

Page 14, line 574: change "On broader…" to "Over broader…"?

Page 14, line 577: "largest month" should be "largest monthly".

Page 15, line 624: should GFAS1.2 also be included as an FRP-based estimation?

Page 16, line 662: change "on inclusion" to "in including".

Page 16, line 675: change "exceeds" to "is greater".

Page 16, line 677: change "emissions is 10%" to "emissions are 10%".

Page 17, lines 713-715: please clarify this last sentence as it isn't clear what is meant "by active fire product". I thought that FINN1.5 and GFED4s are based on the burnt area product available from MODIS.

Page 17, line 721: "scares" should be "scars".

Page 18, line 758: a citation for other model assumptions may be helpful to the reader.

Page 19, final paragraph: while the focus of the evaluation has been based on AOD observations from MODIS and AERONET, it would be useful if some comments could be made on the potential use of in situ, especially aircraft, observations could be used in this context – for example, measurements made during the WE-CAN or FIREX-AQ campaigns in recent years. Also some comment on potential improvements to fire emissions estimates based on FRP products from geostationary satellite observations, especially in combination with low Earth orbit observations such as MODIS (and VIIRS).

Page 30: specify "annual total organic carbon biomass burning emissions"? I also think that removing the sites from the maps could be useful as they aren't that clear to discriminate from the colours on the map, and is a bit distracting from the values in the data.

Page 35, line 1290-1291: clarify that the climatology of AERONET AOD is AERONET-clim in the legend.
* * *

---

## Author Comment (AC1) · 30 Oct 2019

Response to Referee #1

We really appreciate the constructive comments/suggestions from Referee #1, which will greatly help us to improve this manuscript. Following each of the Referee's suggestions <Referee>, we have provided our responses <Response>.

<Referee> Anonymous Referee #1 The authors have run the same atmospheric model with 6 different biomass burning (BB) inventories and analysed the differences using AOD and aeronet. These differences

are often substantial and to some degree the authors have pointed to reasons why those differences exist. I feel the paper helps other modellers in understanding where some of the uncertainties in biomass burning emissions originate from but at the same time I feel the reader is left a bit wondering what the main messages are in the end. Ideally one would come up with recommendations about when and where to use a certain dataset, or when and where to avoid those. But given that the dataset to evaluate the results is also used to construct some this may be too much asked. Please find below a number of suggestions to further improve the paper.

<Response> The six BB datasets analyzed in this study differ in various ways and scales across different biomass burning regions and seasons. Hence, it is challenging to come up with comprehensive recommendations about when and where to use or avoid a particular dataset. Nevertheless, we agree that some recommendations, even in general terms, would be beneficial to the community. Thus, we have added the following statement towards the end of the abstract:

"Although model simulations based on QFED2.4 show overall closest agreement with satellite AOD retrievals, we recommend FEER1.0 for aerosol-focused hindcast experiments in the two biomass-burning dominated regions in the southern hemisphere, SHAF and SHSA (as well as in other regions but with lower confidence), mainly because QFED2.4 is tuned with the GEOS model, whereas FEER1.0 is derived in a more model-independent fashion and is more physical-based since its emission coefficients are independently derived at each grid box." Discussion paper

<Referee>: First sentence in introduction is spelled a bit awkward, please break up in two. Likewise for the second paragraph (L79).

<Response> The first sentence in introduction has been modified to:

Biomass burning (BB) is estimated to contribute about 62% of the global particulate organic carbon (OC) and 27% of black carbon (BC) emissions annually (Wiedinmyer et al., 2011). Therefore, biomass burning emissions significantly affect air quality by acting as a major source of particulate matter (PM), and the climate system by modulating solar radiation and cloud properties.

the second paragraph in introduction has been broken up into:

With the advent of satellite remote sensing of active fire and burned area products in the last couple of decades, a number of global BB emission datasets based on these observations have become available (e.g., Ichoku et al., 2012). Six of such major BB datasets will be compared in this study, including three datasets based on burned area approaches, namely, the Fire INventory from NCAR (FINN, Wiedinmyer et al., 2011) and two versions of the Global Fire Emissions Database (GFED, van der Werf et al., 2006, 2010, 2017), and three datasets based on fire radiative power (FRP) approaches, namely, the Global Fire Assimilation System (GFAS, Kaiser et al., 2012) developed in the European Centre for Medium-Range Weather Forecasts (ECMWF), and two National Aeronautics and Space Administration (NASA) products, i.e., the Fire Energetics and Emissions Research algorithm (FEER, Ichoku and Ellison, 2014) and the Quick Fire Emissions Dataset (QFED, Darmenov and da Silva, 2015).

<Referee> 159: Not sure why that small fire paper is cited in the GFED3 description

<Response> We have removed "Randerson et al., 2012".

<Referee>: 208: Kaiser et al: : :, -> Kaiser et al.,

<Response> This has been corrected.

<Referee> The link on L213 does not work, at least not on my two computers

<Response> They changed the website address recently. Sorry about that. This link in the revised version has been updated as: https://confluence.ecmwf.int/display/CKB/CAMS++Global+Fire+Assimilation+System+%28GFAS%29+data+documentation

<Referee> L282: I am a bit surprised that BB aerosols are injected near the surface. There is quite a bit of literature showing the importance of injection heights in for example the Boreal region

<Response> We agree that this is a concern. Incidentally, this is one of the current limitations of this model and many other models, such as GEOS-chem (Zhu et al., 2018), due to the lack of observational constraint on plume vertical profiles. We have recently promoted an AeroCom multiple-model initiative to constrain the vertical profile of plume height in a model with the MISR plume height (see more details at the Wiki website: https://wiki.met.no/aerocom/phase3-experiments).

<Referee> L297: So basically, you use the same AOD data that was used to construct one of the BB inventories to evaluate a suite of models. That just doesn't feel right and requires careful explanation why this is done and what the consequences are

<Response> We have replaced MODIS AOD with MISRv23 AOD in the Figure 5-7 as below. In general, the results with MISR AOD are consistent with those with MODIS AOD. We also have changed the text part accordingly in the revised version (but too numerous to put here).

<Referee> L305: I feel this is more useful and scientific sound; evaluate the various inventories with independent data

<Response> Please see our response above.

<Referee> L393: But isn't April outside the main fire season in EQAS? In other words, if emissions are very low then a factor two difference (for example due to the detection of small fires in GFED4) is not that noteworthy I guess

<Response> It is actually in August (not April), the peak of the fire season in EQAS, that GFED4s is a factor of two higher than GFED3.1. Sorry about the confusion. We have corrected it in the revised version. Now it reads like this:

"In particular, it is noteworthy that in EQAS, the annual OC emissions from GFED4s was lower than that of GFED3.1 by 18%, but higher by a factor of two in the month of August when peatland burning is predominant."

<Referee> L402: This is indeed a key question and I doubt we will make much progress as long as we keep using one single dataset to constrain emissions. Broadly speaking, the "gas community" (CO, NO2) has shown that the traditional inventories do reasonably well while the "aerosol community" has shown for over 10 years now that the emissions of those inventories are too low to reconstruct measured AOD. It would be very nice if someone would address why those two communities come to different conclusions.

<Response> It is a good point. We have added the statement below in the introduction part:

Andreae (2019) commented that "In contrast to gaseous compounds, which are chemically well defined, aerosols are complex and variable mixtures of organic and inorganic species and comprise particles across a wide range of sizes. This affects in particular the measurements of organic aerosol, black/elemental carbon, and size fractionated aerosol mass".

In the Section 4.3, we mentioned in the manuscript that many models like GEOS version used in this study did not consider the secondary organic carbon produced from biomass burning emissions".

<Referee> L416 lights -> light

<Response> Changed. Thanks.

<Referee> L419: GOES -> GEOS

<Response> Changed. Thanks. Printer-friendly versionDiscussion paper <Referee>: L452: This is a bit confusing, I don't think emissions peaked in April but you found elevated AOD levels due to burning

<Response> This was due to an oversight on our part. Thank you for pointing it out. We rewrote that paragraph as:

[Figure]

"Being mixed with, and often surpassed by other aerosol types in certain regions, however, the contribution of biomass burning aerosols to the total AOD is hardly distinguishable from those of other sources outside of the peak months, such as April (Fig. 6) in the regions of Southeast Asia (SEAS), Central Asia (CEAS), and Boreal Asia (BOAS). Such complicated situations lead to the difficulties in evaluating the BB emission datasets with the AOD observations."

<Referee> L467: Given the very large interannual variability, especially in EQAS, this should be avoided. Please scale with active fire detections or so

<Response> We agree that the biomass burning has large interannual variability in certain regions, especially in EQAS, as we have shown in one of our recent publications on Indonesian fires (Pan et al., 2018). Thus, we overlaid the AERONET climatology and MODIS-Aqua and MODIS-Terra to complement AERONET whenever it has missing data in 2008.

<Referee> L529: Now shown -> Not shown (I guess)

<Response> This has been corrected. Thanks.

<Referee>: L624: This could be a place where this paper could make a difference. Given that the emission factors used in the various datasets are not wildly different, the variability stems from variability in dry matter fuel consumption. GFED has been tuned to match measured fuel consumption, how about FEER and QFAS? Are their levels of fuel consumption (per unit burned area that is) similar to literature-based values? I understand that the FRP approach aims to avoid burned area but these datasets are becoming better constrained and by dividing fuel consumption from FEER and QFED with burned area there could be a useful constraint. Right now we compare AOD with AOD-derived datasets and that just does not help us further I am afraid

<Response> We agree that it would be much more useful to the community to go beyond mere comparisons between the different emissions datasets to develop a constraint that can eventually lead to a realistic understanding of the reasons for the disagreements and how to account for them, and hopefully improve the emissions. The current paper is the initial step toward that goal, as it helps to understand the high-level relationships/disagreements between the different emissions datasets, at the global, regional, and local scales, based on simulations using the exact same global model. Detailed diagnosis of the issues with the individual dataset and finding appropriate synergistic connections between them can follow from this in a systematic manner. Using laboratory measurements of small fires, Ichoku et al. (2008) showed a relationship between the traditional emission factors (EF) based on the burned-biomass approach and the emission coefficients (Ce) based on the FRP approach. These two factors are related via the combustion factor (Fc) that relates time-integrated FRP and total burned biomass. Such relationships can potentially be applied as a useful constraint for improving emissions, but will need to be pursued in a future study that is more focused on addressing such a question.

<Referee> L731: Actually most of the emission factors are from actual fires, not from lab-based measurements.

<Response> We have added the contribution from field campaigns. The paragraph now reads as follows:

"Emission factor (EF). . . .However, the EFs can have significant uncertainties (Andreae, 2019), because each EF deals with a particular experiment or field campaign. Some EFs are derived from lab-based studies whereby samples of fuels are burned in combustion chambers (e.g., Christian et al., 2003; Freeborn et al., 2008), where the combustion characteristics can be very different from those of large-scale open biomass burning and wildfire; and some EFs are derived from field campaigns, where the measurement locations are often not close enough to the biomass burning source due to personnel safety and other logistic factors (Aurell et al., 2019)."

References: Andreae, M. O.: Emission of trace gases and aerosols from biomass

burning – an updated assessment, Atmos. Chem. Phys., 19, 8523–8546, https://doi.org/10.5194/acp-19-8523-2019, 2019.

Aurell, J., Mitchell B., Greenwell D., Holder A., Tabor D., Kiros F., and Gullett B.: Measuring Emission Factors from Open Fires and Detonations. AWMA Air Quality Measurement Methods and Technology, Durham, North Carolina, April 02 - 04, 2019.

Ichoku, C., J. V. Martins, Y. J. Kaufman, M. J. Wooster, P. H. Freeborn, W. M. Hao, S. Baker, C. A. Ryan, and B. L. Nordgren (2008), Laboratory investigation of fire radiative energy and smoke aerosol emissions, J. Geophys. Res., 113, D14S09, doi:10.1029/2007JD009659.

Pan, X., Chin, M., Ichoku, C. M., & Field, R. D. (2018). Connecting Indonesian fires and drought with the type of El Niño and phase of the Indian Ocean dipole during 1979–2016. Journal of Geophysical Research: Atmospheres, 123, 7974–7988. https://doi.org/10.1029/2018JD028402

Zhu, L., M. Val Martin, A. Hecobian, M.N. Deeter, L.V. Gatti, R.A. Kahn, and E.V. Fischer, 2018. Development and implementation of a new biomass burning emissions injection height scheme for the GEOS-Chem model. Geosci. Model Develop. 11, 4103–4116, doi:10.5194/gmd-11-4103-2018.

Please also note the supplement to this comment:
https://www.atmos-chem-phys-discuss.net/acp-2019-475/acp-2019-475-AC1-supplement.pdf

[Figure]

**Fig. 1.** Figure 5. (a) The spatial distribution of monthly mean AOD at 558nm for September 2008 from MISR with the white color representing missing value. The global averaged value (ave) is shown in the parent

[Figure]

[Figure]

**Fig. 2.** Figure 6. Same as Figure 5 except for April 2008.

[Figure]

**Fig. 3.** Figure 7. Monthly variation of AOD (at 550nm wavelength) for 2008 over 14 AERONET sites selected from their respective regions (with its country indicated in parentheses).

---

## Author Comment (AC2) · 30 Oct 2019

Response to Referee #2

We really appreciate the constructive comments from Dr. Parrington. Following each comment/suggestion from Dr. Parrington <Referee>, we have provided our responses below <Response>.

<Referee> The manuscript presents a comparison of biomass burning emissions estimated using satellite observations of active fires including burnt area and fire radiative power. Evaluation of the different emissions datasets is performed by application in a

[Figure]

global aerosol model and comparing the relative changes in the organic matter aerosol fields over MODIS satellite and AERONET ground-based observations of aerosol optical depth (AOD). The authors acknowledge the limitations of the nature of a model-specific study like this but the inter-comparison is very thorough and provides valuable, and timely, insights into variability of estimating biomass burning emissions for application in models. The manuscript is well written and in the scope of Atmospheric Chemistry and Physics, and I recommend it for publication subject to the authors addressing the comments below.

<Response>Thank for your encouraging comment on the merit of this manuscript. We hope that this study will contribute toward advanced understanding of the differences between BB emission datasets, and will eventually facilitate the improvement of the estimation of BB aerosol emissions in models.

General comments:

<Referee> Discussion of uncertainties in emission factors – would the known underestimate of PM emission factors, especially for peat fires in South East Asia, impact on the model AOD? https://www.mdpi.com/2072-4292/10/4/495/htm or https://agupubs.onlinelibrary.wiley.com/doi/pdf/10.1029/2017JD027827

<Response>Yes, it is true that the emission factors estimated from those two studies are far larger than those by Andreae and Merlet (2001) and Akagi (2011). With the higher PM emission factor and thus PM emission, AOD will be enhanced accordingly in the model. In equatorial Asia (EQAS), the experiments based on all six BB emission datasets underestimated AOD during September (the peak of the burn season) to the same degree (∼50%) as the run without any biomass burning emission input, compared to MODIS-Aqua (See Figure 5 and Table S1 in the ACPD version), regardless of whether these BB aerosol emissions are based on the burned-area or FRP approach. This may be largely attributed to missing fire detection from satellite, for example, due to low signal from peat fires, which are predominantly smoldering. In addition, the EF

values for aerosols emitted from peat fires may be underestimated as well (Table 2 in Kiely et al., 2019). Several studies based on in-situ measurements of EF reported that the EFs of PM2.5 for peat fires provided by Andreae and Merlet (2001) and Akagi et al. (2011) as 9.05 and 9.10 g PM2.5 per kg dry matter (see Table 2), respectively, are much lower than their measurements. For example, the studies by Wooster et al. (2018), Stockwell et al. (2016), and Roulston et al. (2018) reported EF values of 21 ± 4.6, 17.8 to 22.3, and 24 g PM2.5 per kg dry matter, respectively, for peat fires in EQAS. Unfortunately, the underestimation of AOD is not shown in the revised version against the MISR AOD, because MISR observation is missing in this region during September 2008. We are asked by the referee #1 to use MISR AOD to evaluate model simulation in the revised version (Figure 5-7), considering QFED and FEER derived their BB emission datasets with historical MODIS AOD.

Specific comments/questions:

<Referee> Page 4, line 107: specify the multi-model study (is it "The AeroCom multi-model study"?).

<Response>Yes, we have clarified it in the revised version as "The AeroCom multi-model study".

<Referee> Page 9, lines 365-367: could it be the case that the two day persistence in FINN1.5 is more representative of peat fires which may be more prevalent in EQAS?

<Response>The two-day persistence approach used by FINN1.5 in the tropical regions may have been more representative of peat fires that are quite prevalent in EQAS. This may be because peat fires typically burn less vigorously and potentially last longer than other fire regimes. We pointed out this in the Section 4.1.2.

<Referee> Page 9, section 3.1.2: it may be useful to describe briefly why 2008 was chosen to investigate the seasonal variation. Does each emissions dataset capture inter-annual variability in the same way?

[Figure]

<Response>The reason we chose 2008 is because it is the year assigned as a benchmark year by AeroCom community with which this study is associated; it is also because the AeroCom Multi-model study of biomass burning lead by Petrenko (mentioned in the introduction part of our manuscript) also chose 2008 as a focus year. As such, the results from these two studies can be intercompared if needed and some synthesized conclusions drawn. In addition, 2008 was chosen because it is a neutral ENSO year, which represents normal burning conditions.

Figure_a shows the comparison of the interannual variation of OC biomass burning emissions in three biomass burning (BB) datasets during the period of 1997-2018 over the Amazon. The three BB emission datasets are FEER, GFED4s, and QFED, which are analyzed in our study. The interannual variability are pronounced across the three BB datasets although with different magnitudes. Apparently, 2007 is the highest burn year, 2009 is the least burn year, while 2008 is a normal burn year. Overall, QFED has the highest OC BB emission, FEER has the second highest, and GFED4s has the least ($\sim$1/3 of QFED) from year to year, which are consistent with our result for 2008. A similar result can be drawn from the region of Africa (Figure_b), where the interannual variability is less pronounced though. In summary, these BB datasets capture similar interannual variability although they have different magnitudes, with QFED having the highest OC BB emission, FEER the second highest, and GFED4s the least.

<Referee> Page 10, line 409: "with each BB emission dataset instead" is repeating the earlier part of the sentence.

<Response>We have changed this sentence to "Therefore, in this study we have implemented all six global BB emission datasets separately in the GEOS model, and evaluated their respective simulated aerosol loadings."

<Referee> Page 11, line 433: change "peaking" to "peak".

<Response>Changed.

<Referee> Page 11, section 3.2.1: it may be useful to a reader to give the names of each region as well as the acronym.

<Response>The full names of the regions have been added in the revised version, such as southern hemisphere South America (SHSA), and southern hemisphere Africa (SHAF).

<Referee> Page 12, section 3.2.2: it may be useful to give the country of the named AERONET sites, which is more intuitive to understanding the geography than giving just the regions.

<Response>The country names of the AERONET sites have been added in Figure 7 as below.

<Referee> Page 12, lines 487-488: "in each respective region".

<Response>We changed to "in each region".

<Referee> Page 12, line 488: change "At most other AERONET" to "At most of the other AERONET".

<Response>Thank you. We have changed "most" to "most of the".

<Referee> Page 13, line 534: "resembled with" should be "resembled".

<Response>We have deleted "with".

<Referee> Page 13, lines 537-538: "All of these evidences" should be "All of this evidence".

<Response> We have changed to "All of this evidence".

<Referee> Page 13, line 539: should "respond" be "correspond".

<Response>The meanings of "respond" and "correspond" are very similar in some sense. Here we prefer to use "respond" to mean doing something in reply.

<Referee> Page 13, line 543 (and other locations): would using "active fire detections" rather than "fire hotspots" be a more scientific way of describing this?

<Response> Changed.

<Referee> Page 13, line 553: "over entire" should be "over the entire".

<Response> Changed.

<Referee> Page 14, line 566: should "emitted from smoke aerosols" be "emitted as smoke aerosols"?

<Response> Changed to "the dominance of the fine-mode aerosol particles in smoke aerosols".

<Referee> Page 14, line 567: change "These evidences" to "This".

<Response> We changed to "This evidence".

<Referee> Page 14, line 574: change "On broader: : :" to "Over broader: : :"?

<Response> Changed to "in regional emission", which is relative to the local scale.

<Referee> Page 14, line 577: "largest month" should be "largest monthly".

<Response>Changed.

<Referee> Page 15, line 624: should GFAS1.2 also be included as an FRP-based estimation?

<Response> Right, we have added GFAS1.2.

<Referee> Page 16, line 662: change "on inclusion" to "in including".

<Response> Changed "on inclusion of" to "in including".

<Referee> Page 16, line 675: change "exceeds" to "is greater".

<Response> Changed.

<Referee> Page 16, line 677: change "emissions is 10%" to "emissions are 10%".

<Response> Changed.

<Referee> Page 17, lines 713-715: please clarify this last sentence as it isn't clear what is meant "by active fire product". I thought that FINN1.5 and GFED4s are based on the burnt area product available from MODIS.

<Response> FINN1.5 actually uses active fire product to estimate the burned area by assuming each active fire pixel represents a burned area of 1 km2 for most biome types (see details in Sect. 2.1.3). GFED4s uses the official burned area product for large fires, but estimates burned area for small fires using active fire detections. We have rewritten the sentences in the revised version as:

"This issue also affects FINN1.5 (Wiedinmyer et al., 2011), which derives the burned area by assuming each active fire pixel to correspond to a burned area of 1 km2 for most biome types (see details in Sect. 2.1.3), and GFED4s, which uses burned area product for large fires but derives burned areas for small fires using the MODIS active fire product."

<Referee> Page 17, line 721: "scares" should be "scars".

<Response> Changed. Printer-friendly pper

<Referee> Page 18, line 758: a citation for other model assumptions may be helpful to the reader.

<Response> We have removed this vague expression, i.e., other model assumptions, from the revised version.

<Referee> Page 19, final paragraph: while the focus of the evaluation has been based on AOD observations from MODIS and AERONET, it would be useful if some comments could be made on the potential use of in situ, especially aircraft, observations could be used in this context – for example, measurements made during the WE-CAN

or FIREXAQ campaigns in recent years. Also some comment on potential improve-
ments to fire emissions estimates based on FRP products from geostationary satel-
lite observations, especially in combination with low Earth orbit observations such as
MODIS (and VIIRS).

<Response> Thank you for your suggestions. We have added your suggestions in the
last paragraph as

"The investigated global BB emission datasets driven by fire remote sensing and
retrievals of FRP and burned-area products, which have hitherto depended heavily
on MODIS, can be augmented with products from higher resolution sensors such
as Visible Infrared Imaging Radiometer Suite (VIIRS), and the global suite of geo-
stationary meteorological satellites such as Meteosat (covering Europe, Africa and
the Indian Ocean), Geostationary Operational Environmental Satellite (GOES, cov-
ering North, Central, and South America) and Himawari (covering east Asia, south-
east Asia, and Australia). Also, measurements from the recent field campaigns
such as WE-CAN (https://www.eol.ucar.edu/field_projects/we-can) and FIREX-AQ
(https://www.esrl.noaa.gov/csd/projects/firex-aq/science/motivation.html) are expected
to contribute toward advancing our knowledge of biomass burning emissions in North
America. The evaluation in this study has been solely based on remote sensing
AOD data, including retrievals from both satellite and ground-based (AERONET) sen-
sors. Continuous mass concentration measurements are needed to validate the fire-
generated aerosol loading in specific contexts, such as in analyzing collocated surface
and vertical aerosol concentrations and composition, at least in the major BB regions."

<Referee> Page 30: specify "annual total organic carbon biomass burning emissions"?
I also think that removing the sites from the maps could be useful as they aren't that
clear to discriminate from the colours on the map, and is a bit distracting from the
values in the data.

<Response> We have specified "The spatial distribution of annual total organic carbon

biomass burning emissions" in the caption of Figure 2. The AERONET sites have been removed from Figure 2 as below.

<Referee> Page 35, line 1290-1291: clarify that the climatology of AERONET AOD is AERONETclim in the legend. <Response> We have added it in the caption of Figure 7, "The climatology of AERONET AOD (i.e., AERONET-clim)".

References: Kiely, L., Spracklen, D. V., Wiedinmyer, C., Conibear, L., Reddington, C. L., Archer-Nicholls, S., Lowe, D., Arnold, S. R., Knote, C., Khan, M. F., Latif, M. T., Kuwata, M., Budisulistiorini, S. H., and Syaufina, L.: New estimate of particulate emissions from Indonesian peat fires in 2015, Atmos. Chem. Phys., 19, 11105–11121, https://doi.org/10.5194/acp-19-11105-2019, 2019.

Roulston, C., Paton-Walsh, C., Smith, T. E. L., Guérette, É.-A., Evers, S., Yule, C. M., et al.: Fine particle emissions from tropical peat fires decrease rapidly with time since ignition. Journal of Geophysical Research: Atmospheres, 123, 5607–5617. https://doi.org/10.1029/2017JD027827, 2018.

Stockwell, C.E., Jayarathne, T., Cochrane, M.A., Ryan, K.C., Putra, E.I., Saharjo, B.H., Nurhayati, A.D., Albar, I., Blake, D.R., Simpson, I.J., et al.: Field measurements of trace gases and aerosols emitted by peat fires in Central Kalimantan, Indonesia during the 2015 El Niño. Atmos. Chem. Phys., 16, 11711–11732, 2016.

Wooster, M., Gaveau, D., Salim, M., Zhang, T., Xu, W., Green, D., Huijnen, V., Murdiyarso, D., Gunawan, D., Borchard, N., Schirrmann, M., Main, B. and Sepriando, A.:New Tropical Peatland Gas and Particulate Emissions Factors Indicate 2015 Indonesian Fires Released Far More Particulate Matter (but Less Methane) than Current Inventories Imply. Remote Sensing. 10 (4), p.495, https://doi.org/10.3390/rs10040495, 2018.

[Figure]

[Figure]

OC biomass burning emission for 2008

(a) GFED3.1 (sum=15.65 Tg yr⁻¹)  (b) GFED4s (sum=13.76 Tg yr⁻¹)

(c) FINN1.5 (sum=19.48 Tg yr⁻¹)  (d) GFAS1.2 (sum=18.22 Tg yr⁻¹)

(e) FEER1.0 (sum=28.48 Tg yr⁻¹)  (f) QFED2.4 (sum=51.93 Tg yr⁻¹)

(Units:g m⁻² yr⁻¹)

0   0.001   0.10   0.20   0.30   0.50   1.0   2.0   3.0   5.0   7.0   38

**Fig. 1.** Figure 2. The spatial distribution of annual total organic carbon biomass burning emissions for 2008 estimated by six biomass burning emission datasets (units: g m-2 yr-1).

[Figure]

**Fig. 2.** Figure 7. Monthly variation of AOD (at 550nm wavelength) for 2008 over 14 AERONET sites selected from their respective regions (with its country indicated in parentheses).

[Figure]

**Fig. 3.** Figure_a. The comparison of the interannual variation of OC biomass burning emissions in three biomass burning datasets during the period of 1997-2018 over Amazon (80W-30W, 60S-15N).

[Figure]

**Fig. 4.** Figure_b. The comparison of the interannual variation of OC biomass burning emissions in three biomass burning datasets during the period of 1997-2018 over Africa (24W-50E, 40S-20N).

---

## Author Response (AR1)

**Response to Referee #1**

We really appreciate the constructive comments/suggestions from Referee #1, which will greatly help us to improve this manuscript. We have provided our responses in blue-colored font following each of the Referee's suggestions (below).

Anonymous Referee #1

The authors have run the same atmospheric model with 6 different biomass burning (BB) inventories and analysed the differences using AOD and aeronet. These differences are often substantial and to some degree the authors have pointed to reasons why those differences exist. I feel the paper helps other modellers in understanding where some of the uncertainties in biomass burning emissions originate from but at the same time I feel the reader is left a bit wondering what t**he main messages are in the end**. Ideally one would come up with recommendations about when and where to use a certain dataset, or when and where to avoid those. But given that the dataset to evaluate the results is also used to construct some this may be too much asked. Please find below a number of suggestions to further improve the paper.
**Response:** The six BB datasets analyzed in this study differ in various ways and scales across different biomass burning regions and seasons. Hence, it is challenging to come up with comprehensive recommendations about when and where to use or avoid a particular dataset. Nevertheless, we agree that some recommendations, even in general terms, would be beneficial to the community. Thus, we have added the following statement towards the end of the abstract:

"Although model simulations based on QFED2.4 show overall closest agreement with satellite AOD retrievals, we recommend FEER1.0 for aerosol-focused hindcast experiments in the two biomass-burning dominated regions in the southern hemisphere, SHAF and SHSA (as well as in other regions but with lower confidence), mainly because QFED2.4 is tuned with the GEOS model, whereas FEER1.0 is derived in a more model-independent fashion and is more physical-based since its emission coefficients are independently derived at each grid box."

First sentence in introduction is spelled a bit awkward, please break up in two. Likewise for the second paragraph (L79).
**Response:** The first sentence in introduction has been modified to:

Biomass burning (BB) is estimated to contribute about 62% of the global particulate organic carbon (OC) and 27% of black carbon (BC) emissions annually (Wiedinmyer et al., 2011). Therefore, biomass burning emissions significantly affect air quality by acting as a major source of particulate matter (PM), and the climate system by modulating solar radiation and cloud properties.

the second paragraph in introduction has been broken up into:

With the advent of satellite remote sensing of active fire and burned area products in the last couple of decades, a number of global BB emission datasets based on these observations have become available (e.g., Ichoku et al., 2012). Six of such major BB datasets will be compared in this study, including three datasets based on burned area approaches, namely, the Fire INventory from NCAR (FINN, Wiedinmyer et al., 2011) and two versions of the Global Fire Emissions Database (GFED, van der Werf et al., 2006, 2010, 2017), and three datasets based on fire radiative power (FRP) approaches, namely, the Global Fire Assimilation System (GFAS, Kaiser et al., 2012) developed in the European Centre for Medium-Range Weather Forecasts (ECMWF), and two National Aeronautics and Space Administration (NASA) products, i.e., the Fire Energetics and Emissions Research algorithm (FEER, Ichoku and Ellison, 2014) and the Quick Fire Emissions Dataset (QFED, Darmenov and da Silva, 2015).

159: Not sure why that small fire paper is cited in the GFED3 description
**Response:** We have removed "Randerson et al., 2012".

208: Kaiser et al: : :, -> Kaiser et al.,
**Response:** This has been corrected.

The link on L213 does not work, at least not on my two computers
**Response:** They changed the website address recently. Sorry about that. This link in the revised version has been updated as:
https://confluence.ecmwf.int/display/CKB/CAMS++Global+Fire+Assimilation+System+%28GFAS%29+data+documentation

L282: I am a bit surprised that BB aerosols are injected near the surface. There is quite a bit of literature showing the importance of injection heights in for example the Boreal region
**Response:** We agree that this is a concern. Incidentally, this is one of the current limitations of this model and many other models, such as GEOS-chem (Zhu et al., 2018), due to the lack of observational constraint on plume vertical profiles. We have recently promoted an AeroCom multiple-model initiative to constrain the vertical profile of plume height in a model with the MISR plume height (see more details at the Wiki website: https://wiki.met.no/aerocom/phase3-experiments).

L297: So basically, you use the same AOD data that was used to construct one of the BB inventories to evaluate a suite of models. That just doesn't feel right and requires careful explanation why this is done and what the consequences are
**Response:** We have replaced MODIS AOD with MISRv23 AOD in the Figure 5-7 as below. In general, the results with MISR AOD are consistent with those with MODIS AOD. We also have changed the text part accordingly in the revised version (but not shown here because too numerous).

[Figure]

**AOD (550nm) for Apr 2008**

[Figure]

[Figure]

Monthly AOD (550nm) at AERONET sites for 2008

L305: I feel this is more useful and scientific sound; evaluate the various inventories with independent data
**Response:** Please see our response above.

L393: But isn't April outside the main fire season in EQAS? In other words, if emissions are very low then a factor two difference (for example due to the detection of small fires in GFED4) is not that noteworthy I guess
**Response:** It is actually in August (not April), the peak of the fire season in EQAS, that GFED4s is a factor of two higher than GFED3.1. Sorry about the confusion. We have corrected it in the revised version. Now it reads like this:

"In particular, it is noteworthy that in EQAS, the annual OC emissions from GFED4s was lower than that of GFED3.1 by 18%, but higher by a factor of two in the month of **August** when peatland burning is predominant."

L402: This is indeed a key question and I doubt we will make much progress as long as we keep using one single dataset to constrain emissions. Broadly speaking, the "gas community" (CO, NO2) has shown that the traditional inventories do reasonably well while the "aerosol community" has shown for over 10 years now that the emissions of those inventories are too low to reconstruct measured AOD. It would be very nice if someone would address why those two communities come to different conclusions.
**Response:** It is a good point. We have added the statement below in the introduction part:

Andreae (2019) commented that "In contrast to gaseous compounds, which are chemically well defined, aerosols are complex and variable mixtures of organic and inorganic species and comprise particles across a wide range of sizes. This affects in particular the measurements of organic aerosol, black/elemental carbon, and size fractionated aerosol mass".

In the Section 4.3, we mentioned in the manuscript that many models like GEOS version used in this study did not consider the secondary organic carbon produced from biomass burning emissions".

L416 lights -> light
**Response:** Changed. Thanks.

L419: GOES -> GEOS
**Response:** Changed. Thanks.

L452: This is a bit confusing, I don't think emissions peaked in April but you found elevated AOD levels due to burning
**Response:** This was due to an oversight on our part. Thank you for pointing it out. We rewrote that paragraph as:

"Being mixed with, and often surpassed by, other aerosol types in certain regions, however, the contribution of biomass burning aerosols to the total AOD is hardly distinguishable from those of other sources in the peak months, such as April (Fig. 6) in the regions of Southeast Asia (SEAS), Central Asia (CEAS), and Boreal Asia (BOAS)."

L467: Given the very large interannual variability, especially in EQAS, this should be avoided. Please scale with active fire detections or so
**Response:** We agree that the biomass burning has large interannual variability in certain regions, especially in EQAS, as we have shown in one of our recent publications on Indonesian fires (Pan et al., 2018). Thus, we overlaid the AERONET climatology and MODIS-Aqua and MODIS-Terra to complement AERONET whenever it has missing data in 2008.

L529: Now shown -> Not shown (I guess)
**Response:** This has been corrected. Thanks.

L624: This could be a place where this paper could make a difference. Given that the emission factors used in the various datasets are not wildly different, the variability stems from variability in dry matter fuel consumption. GFED has been tuned to match measured fuel consumption, how about FEER and QFAS? Are their levels of fuel consumption (per unit burned area that is) similar to literature-based values? I understand that the FRP approach aims to avoid burned area but these datasets are becoming better constrained and by dividing fuel consumption from FEER and QFED with burned area there could be a useful constraint. Right now we compare AOD with AOD-derived datasets and that just does not help us further I am afraid

**Response:** We agree that it would be much more useful to the community to go beyond mere comparisons between the different emissions datasets to develop a constraint that can eventually lead to a realistic understanding of the reasons for the disagreements and how to account for them, and hopefully improve the emissions. The current paper is the initial step toward that goal, as it helps to understand the high-level relationships/disagreements between the different emissions datasets, at the global, regional, and local scales, based on simulations using the exact same global model. Detailed diagnosis of the issues with the individual dataset and finding appropriate synergistic connections between them can follow from this in a systematic manner. Using laboratory measurements of small fires, Ichoku et al. (2008) showed a relationship between the traditional emission factors (EF) based on the burned-biomass approach and the emission coefficients (Ce) based on the FRP approach. These two factors are related via the combustion factor (Fc) that relates time-integrated FRP and total burned biomass. Such relationships can potentially be applied as a useful constraint for improving emissions, but will need to be pursued in a future study that is more focused on addressing such a question.

L731: Actually most of the emission factors are from actual fires, not from lab-based measurements.

**Response:** We have added the contribution from field campaigns. The paragraph now reads as follows:

"*Emission factor (EF).* ... However, the EFs can have significant uncertainties (Andreae, 2019), because each EF results from a particular experiment or field campaign. Some *EF*s are derived from lab-based studies whereby samples of fuels are burned in combustion chambers (e.g., Christian et al., 2003; Freeborn et al., 2008), where the combustion characteristics can be very different from those of large-scale open biomass burning and wildfires; and some *EFs* are derived from field campaigns, where the measurement locations are often not close enough to the biomass burning source due to personnel safety and other logistic factors (Aurell et al., 2019)."

Page 10, line 409: "with each BB emission dataset instead" is repeating the earlier part of the sentence.

**Response:** We have changed this sentence to "Therefore, in this study we have implemented all six global BB emission datasets separately in the GEOS model, and evaluated their respective simulated aerosol loadings."

Page 11, line 433: change "peaking" to "peak".

**Response:** Changed.

Page 11, section 3.2.1: it may be useful to a reader to give the names of each region as well as the acronym.

**Response:** The full names of the regions have been added in the revised version, such as southern hemisphere South America (SHSA), and southern hemisphere Africa (SHAF).

Page 12, section 3.2.2: it may be useful to give the country of the named AERONET sites, which is more intuitive to understanding the geography than giving just the regions.

**Response:** The country names of the AERONET sites have been added in Figure 7 as below.

[Figure]

Page 12, lines 487-488: "in each respective region".
**Response:** We changed to "in each region".
Page 12, line 488: change "At most other AERONET" to "At most of the other AERONET".
**Response:** Thank you. We have changed "most" to "most of the".
Page 13, line 534: "resembled with" should be "resembled".
**Response:** We have deleted "with".
Page 13, lines 537-538: "All of these evidences" should be "All of this evidence".
**Response:** We have changed to "All of this evidence".
Page 13, line 539: should "respond" be "correspond".
**Response:** The meanings of "respond" and "correspond" are very similar in some sense. Here we prefer to use "respond" to mean doing something in reply.
Page 13, line 543 (and other locations): would using "active fire detections" rather than "fire hotspots" be a more scientific way of describing this?
**Response:** Changed.
Page 13, line 553: "over entire" should be "over the entire".
**Response:** Changed.
Page 14, line 566: should "emitted from smoke aerosols" be "emitted as smoke aerosols"?
**Response:** Changed to "the dominance of the fine-mode aerosol particles in smoke aerosols".
Page 14, line 567: change "These evidences" to "This".
**Response:** We changed to "This evidence".
Page 14, line 574: change "On broader: : :" to "Over broader: : :"?
**Response:** Changed to "in regional emission", which is relative to the local scale.
Page 14, line 577: "largest month" should be "largest monthly".
**Response:** Changed.
Page 15, line 624: should GFAS1.2 also be included as an FRP-based estimation?
**Response:** Right, we have added GFAS1.2.
Page 16, line 662: change "on inclusion" to "in including".
**Response:** Changed "on inclusion of" to "in including".
Page 16, line 675: change "exceeds" to "is greater".
**Response:** Changed.
Page 16, line 677: change "emissions is 10%" to "emissions are 10%".
**Response:** Changed.
Page 17, lines 713-715: please clarify this last sentence as it isn't clear what is meant "by active fire product". I thought that FINN1.5 and GFED4s are based on the burnt area product available from MODIS.
**Response:** FINN1.5 actually uses active fire product to estimate the burned area by assuming each active fire pixel represents a burned area of 1 km$^2$ for most biome types (see details in *Sect.* 2.1.3). GFED4s uses the official burned area product for large fires, but estimates burned area for small fires using active fire detections. We have rewritten the sentences in the revised version as:

"This issue also affects FINN1.5 (Wiedinmyer et al., 2011), which derives the burned area by assuming each active fire pixel to correspond to a burned area of 1 km$^2$ for most biome types (see details in *Sect.* 2.1.3), and GFED4s, which uses burned area product for large fires but derives burned areas for small fires using the MODIS active fire product."

Page 17, line 721: "scares" should be "scars".
**Response:** Changed.
Page 18, line 758: a citation for other model assumptions may be helpful to the reader.
**Response:** We have removed this vague expression, i.e., other model assumptions, from the revised version.
Page 19, final paragraph: while the focus of the evaluation has been based on AOD observations from MODIS and AERONET, it would be useful if some comments could be made on the potential use of in situ, especially aircraft, observations could be used in this context – for example, measurements made during the WE-CAN or FIREXAQ campaigns in recent years. Also some comment on potential improvements to fire emissions estimates based on FRP products from geostationary satellite observations, especially in combination with low Earth orbit observations such as MODIS (and VIIRS).
**Response:** Thank you for your suggestions. We have added your suggestions in the last paragraph as

"The investigated global BB emission datasets driven by fire remote sensing and retrievals of FRP and burned-area products, which have hitherto depended heavily on MODIS, can be augmented with products from higher resolution sensors such as Visible Infrared Imaging Radiometer Suite (VIIRS), and the global suite of geostationary meteorological satellites such as Meteosat (covering Europe, Africa and the Indian Ocean), Geostationary Operational Environmental Satellite (GOES, covering North, Central, and South America) and Himawari (covering east Asia, southeast Asia, and Australia). Also, measurements from the recent field campaigns such as WE-CAN (https://www.eol.ucar.edu/field_projects/we-can) and FIREX-AQ (https://www.esrl.noaa.gov/csd/projects/firex-aq/science/motivation.html) are expected to contribute toward advancing our knowledge of biomass burning emissions in North America. The evaluation in this study has been solely based on remote sensing AOD data, including retrievals from both satellite and ground-based (AERONET) sensors. Continuous mass concentration measurements are needed to validate the fire-generated aerosol loading in specific contexts, such as in analyzing collocated surface and vertical aerosol concentrations and composition, at least in the major BB regions."

Page 30: specify "annual total organic carbon biomass burning emissions"? I also think that removing the sites from the maps could be useful as they aren't that clear to discriminate from the colours on the map, and is a bit distracting from the values in the data.
**Response:** We have specified "The spatial distribution of annual total organic carbon biomass burning emissions" in the caption of Figure 2. The AERONET sites have been removed from Figure 2 as below.

[Figure]

OC biomass burning emission for 2008

(a) GFED3.1 (sum=15.65 Tg yr$^{-1}$)
(b) GFED4s (sum=13.76 Tg yr$^{-1}$)
(c) FINN1.5 (sum=19.48 Tg yr$^{-1}$)
(d) GFAS1.2 (sum=18.22 Tg yr$^{-1}$)
(e) FEER1.0 (sum=28.48 Tg yr$^{-1}$)
(f) QFED2.4 (sum=51.93 Tg yr$^{-1}$)

(Units:g m$^{-2}$ yr$^{-1}$)

Page 35, line 1290-1291: clarify that the climatology of AERONET AOD is AERONETclim in the legend.
**Response:** We have added it in the caption of Figure 7, "The climatology of AERONET AOD (i.e., AERONET-clim)".

(a) GFED3.1 (sum=1.76 Tg yr⁻¹)   (b) GFED4s (sum=1.65 Tg yr⁻¹)
(c) FINN1.5 (sum=1.83 Tg yr⁻¹)   (d) GFAS1.2 (sum=1.99 Tg yr⁻¹)
(e) FEER1.0 (sum=3.66 Tg yr⁻¹)   (f) QFED2.4 (sum=5.54 Tg yr⁻¹)

(Units:g m⁻² yr⁻¹)

Figure S1. The spatial distribution of annual total black carbon biomass burning emissions for
2008 estimated by six biomass burning emission datasets. The global total amount is indicated in
the parentheses. The fourteen selected AERONET sites are labeled as magenta dots.

[Figure]

BC biomass burning emission for 2008

| | BONA | TENA | CEAM | NHSA | SHSA | EURO | MIDE | NHAF | SHAF | BOAS | CEAS | SEAS | EQAS | AUST |
|---|---|---|---|---|---|---|---|---|---|---|---|---|---|---|
| mean | 0.05 | 0.07 | 0.06 | 0.04 | 0.39 | 0.01 | 0.01 | 0.55 | 0.80 | 0.26 | 0.16 | 0.18 | 0.04 | 0.10 |
| std | 0.05 | 0.08 | 0.04 | 0.03 | 0.22 | 0.01 | 0.02 | 0.28 | 0.46 | 0.19 | 0.11 | 0.12 | 0.02 | 0.08 |
| max/min | 18.45 | 16.54 | 8.05 | 5.77 | 4.18 | 11.83 | 57.96 | 3.54 | 3.97 | 6.48 | 7.00 | 4.98 | 3.56 | 18.95 |
| cv | 1.00 | 1.22 | 0.70 | 0.66 | 0.55 | 0.78 | 1.53 | 0.50 | 0.57 | 0.73 | 0.71 | 0.66 | 0.49 | 0.78 |
| rank of cv | 12 | 13 | 7 | 6 | 3 | 11 | 14 | 2 | 4 | 9 | 8 | 5 | 1 | 10 |

| Legend | GLOB_TOT |
|---|---|
| GFED3.1 | 1.76 |
| GFED4s | 1.65 |
| FINN1.5 | 1.83 |
| GFAS1.2 | 1.99 |
| FEER1.0 | 3.66 |
| QFED2.4 | 5.54 |

Figure S2. The regional annual total black carbon biomass burning emissions for 2008 in six
biomass burning emission datasets in 14 regions (units: Tg yr⁻¹). The global annual total amount
is listed after the name of each dataset (GLOB_TOT). Relevant statistics for the six BB emission
datasets in each region are also listed under the short name of this region on the top of the panel
in blue, with the mean of six BB emission datasets in the first row. Three different methods to
measure the dispersion of the six BB emission datasets are shown as well: one absolute method,
i.e., the standard deviation (std) in the second row, and two relative methods, i.e., the ratio of max
(maximum) to min (minimum) shown in the third row, and the coefficient of variation (cv),
defined as the ratio of the std to the mean, in the fourth row. The rankings of the regions
regarding the dispersion of the BB emissions datasets according to cv are shown in the fifth row
(i.e., a ranking of 1 means that this region shows the least spread among the six BB emissions
datasets, while a ranking of 14 indicates this region has the largest spread in the 14 regions).

[Figure]

Figure S3. Monthly variation of black carbon biomass burning emissions for 2008 in six biomass burning emission datasets in 14 regions and the global (GLOB, highlighted with a black box). The annual total emission is listed in the right side of each panel.

Table S1. The regional averaged monthly mean AOD at 550nm from MISR along with model biases (i.e., model minus MISR) in seven model experiments, i.e., NOBB, GFED3.1, GFED4s, FINN1.5, GFAS1.2, FEER1.0, and QFED2.4, for September and April 2008, respectively.

| Dataset | BONA | TENA | CEAM | NHSA | SHSA | EURO | MIDE | NHAF | SHAF | BOAS | CEAS | SEAS | EQAS | AUST | GLOB |
|---|---|---|---|---|---|---|---|---|---|---|---|---|---|---|---|
| | | | | | | September 2008 | | | | | | | | | |
| MISR | 0.127 | 0.129 | 0.133 | 0.141 | 0.188 | 0.139 | 0.334 | 0.411 | 0.331 | 0.135 | 0.219 | 0.257 | 0.138 | 0.085 | 0.218 |
| NOBB | 0.04 | 0.011 | -0.031* | -0.083 | -0.132 | 0.084 | -0.013 | -0.05 | -0.283 | 0.043 | 0.053 | 0.025 | -0.047 | -0.031 | -0.029 |
| GFED3.1 | 0.042 | 0.013 | -0.031 | -0.074 | -0.078 | 0.085 | -0.012 | -0.046 | -0.178 | 0.048 | 0.055 | 0.026 | -0.039 | -0.02 | -0.012 |
| GFED4s | 0.044 | 0.014 | -0.031 | -0.075 | -0.081 | 0.085 | -0.012 | -0.047 | -0.208 | 0.047 | 0.056 | 0.025 | -0.04 | -0.023 | -0.015 |
| FINN1.5 | 0.041 | 0.013 | -0.031 | -0.07 | -0.04 | 0.08 | -0.013 | -0.047 | -0.21 | 0.045 | 0.05 | 0.016 | -0.033 | -0.025 | -0.012 |
| GFAS1.2 | 0.042 | 0.013 | -0.031 | -0.074 | -0.08 | 0.08 | -0.013 | -0.047 | -0.208 | 0.046 | 0.05 | 0.016 | -0.029 | -0.02 | -0.016 |
| FEER1.0 | 0.044 | 0.016 | -0.03 | -0.061 | -0.021 | 0.088 | -0.009 | -0.039 | -0.079 | 0.049 | 0.061 | 0.027 | -0.032 | -0.01 | 0.006 |
| QFED2.4 | 0.058 | 0.04 | -0.023 | -0.058 | 0.02 | 0.097 | -0.004 | -0.035 | -0.044 | 0.058 | 0.068 | 0.028 | -0.029 | -0.002 | 0.02 |
| | | | | | | April 2008 | | | | | | | | | |
| MISR | 0.192 | 0.16 | 0.182 | 0.207 | 0.067 | 0.148 | 0.381 | 0.446 | 0.096 | 0.221 | 0.324 | 0.363 | 0.118 | 0.049 | 0.242 |
| NOBB | -0.041 | -0.016 | -0.083 | -0.086 | -0.016 | 0.077 | -0.059 | -0.082 | -0.042 | -0.005 | 0.017 | -0.074 | -0.028 | -0.004 | -0.029 |
| GFED3.1 | -0.023 | -0.008 | -0.077 | -0.082 | -0.014 | 0.079 | -0.058 | -0.078 | -0.04 | 0.104 | 0.055 | -0.057 | -0.027 | -0.002 | -0.015 |
| GFED4s | -0.026 | -0.008 | -0.076 | -0.081 | -0.015 | 0.081 | -0.057 | -0.079 | -0.039 | 0.095 | 0.05 | -0.063 | -0.027 | -0.002 | -0.016 |
| FINN1.5 | -0.031 | -0.006 | -0.056 | -0.072 | -0.013 | 0.075 | -0.055 | -0.068 | -0.037 | 0.041 | 0.069 | 0.014 | -0.024 | -0.002 | -0.008 |
| GFAS1.2 | -0.014 | -0.005 | -0.07 | -0.075 | -0.014 | 0.073 | -0.057 | -0.073 | -0.039 | 0.164 | 0.07 | -0.058 | -0.025 | -0.001 | -0.008 |
| FEER1.0 | -0.017 | 0.003 | -0.054 | -0.059 | -0.011 | 0.089 | -0.053 | -0.068 | -0.036 | 0.162 | 0.083 | -0.028 | -0.024 | 0 | -0.001 |
| QFED2.4 | 0.018 | 0.027 | -0.026 | -0.054 | -0.008 | 0.099 | -0.047 | -0.059 | -0.032 | 0.329 | 0.17 | 0.038 | -0.022 | 0.004 | 0.031 |

*Highlighted in blue if negative bias.

Deleted: Table S1. The regional averaged monthly mean AOD at 550nm in MODIS-Aqua (i.e., MOD-a) along with model biases (i.e., model minus MODIS-a) in seven model experiments, i.e., NOBB, GFED3.1, GFED4s, FINN1.5, GFAS1.2, FEER1.0, and QFED2.4, for September and April 2008, respectively. ¶
Dataset¶
... [51]

| Page 12: [1] Deleted | Xiaohua Pan | 10/7/19 1:06:00 PM |
|---|---|---|

| Page 12: [1] Deleted | Xiaohua Pan | 10/7/19 1:06:00 PM |
|---|---|---|

| Page 12: [1] Deleted | Xiaohua Pan | 10/7/19 1:06:00 PM |
|---|---|---|

| Page 12: [2] Formatted | Xiaohua Pan | 10/30/19 2:14:00 PM |
|---|---|---|

Font color: Text 1

| Page 12: [2] Formatted | Xiaohua Pan | 10/30/19 2:14:00 PM |
|---|---|---|

Font color: Text 1

| Page 12: [2] Formatted | Xiaohua Pan | 10/30/19 2:14:00 PM |
|---|---|---|

Font color: Text 1

| Page 12: [3] Deleted | Xiaohua Pan | 10/7/19 1:14:00 PM |
|---|---|---|

| Page 12: [3] Deleted | Xiaohua Pan | 10/7/19 1:14:00 PM |
|---|---|---|

| Page 12: [3] Deleted | Xiaohua Pan | 10/7/19 1:14:00 PM |
|---|---|---|

| Page 12: [4] Formatted | Xiaohua Pan | 10/30/19 2:14:00 PM |
|---|---|---|

Font color: Text 1

| Page 12: [4] Formatted | Xiaohua Pan | 10/30/19 2:14:00 PM |
|---|---|---|

Font color: Text 1

| Page 12: [5] Deleted | Ichoku, Charles | 10/26/19 11:43:00 AM |
|---|---|---|

| Page 12: [5] Deleted | Ichoku, Charles | 10/26/19 11:43:00 AM |
|---|---|---|

| Page 12: [6] Deleted | Xiaohua Pan | 10/28/19 4:34:00 PM |
|---|---|---|

| Page 12: [6] Deleted | Xiaohua Pan | 10/28/19 4:34:00 PM |
|---|---|---|

| Page 12: [6] Deleted | Xiaohua Pan | 10/28/19 4:34:00 PM |
|---|---|---|

| Page 12: [6] Deleted | Xiaohua Pan | 10/28/19 4:34:00 PM |
|---|---|---|

| Page 12: [6] Deleted | Xiaohua Pan | 10/28/19 4:34:00 PM |
|---|---|---|

| Page 12: [6] Deleted | Xiaohua Pan | 10/28/19 4:34:00 PM |
|---|---|---|

| Page 12: [7] Deleted | Xiaohua Pan | 10/11/19 10:41:00 AM |
|---|---|---|

| Page 12: [7] Deleted | Xiaohua Pan | 10/11/19 10:41:00 AM |
|---|---|---|

| Page 12: [8] Deleted | Xiaohua Pan | 10/11/19 10:50:00 AM |
|---|---|---|

| Page 12: [8] Deleted | Xiaohua Pan | 10/11/19 10:50:00 AM |
|---|---|---|

| Page 12: [8] Deleted | Xiaohua Pan | 10/11/19 10:50:00 AM |
|---|---|---|

| Page 12: [8] Deleted | Xiaohua Pan | 10/11/19 10:50:00 AM |
|---|---|---|

| Page 12: [8] Deleted | Xiaohua Pan | 10/11/19 10:50:00 AM |
|---|---|---|

| Page 12: [9] Deleted | Xiaohua Pan | 10/11/19 10:56:00 AM |
|---|---|---|

| Page 12: [9] Deleted | Xiaohua Pan | 10/11/19 10:56:00 AM |
|---|---|---|

| Page 12: [9] Deleted | Xiaohua Pan | 10/11/19 10:56:00 AM |
|---|---|---|

| Page 12: [9] Deleted | Xiaohua Pan | 10/11/19 10:56:00 AM |
|---|---|---|

| Page 12: [9] Deleted | Xiaohua Pan | 10/11/19 10:56:00 AM |
|---|---|---|

| Page 12: [10] Formatted | Xiaohua Pan | 10/30/19 2:14:00 PM |
|---|---|---|

Font color: Text 1

| Page 12: [10] Formatted | Xiaohua Pan | 10/30/19 2:14:00 PM |
|---|---|---|

Font color: Text 1

| Page 12: [11] Deleted | Xiaohua Pan | 10/11/19 10:59:00 AM |
|---|---|---|

| Page 12: [11] Deleted | Xiaohua Pan | 10/11/19 10:59:00 AM |
|---|---|---|

| Page 13: [12] Deleted | Xiaohua Pan | 10/31/19 2:14:00 PM |
|---|---|---|

| Page 13: [12] Deleted | Xiaohua Pan | 10/31/19 2:14:00 PM |
|---|---|---|

| Page 13: [13] Deleted | Ichoku, Charles | 10/26/19 11:54:00 AM |

| Page 13: [13] Deleted | Ichoku, Charles | 10/26/19 11:54:00 AM |

| Page 13: [14] Deleted | Xiaohua Pan | 10/31/19 2:23:00 PM |

| Page 13: [14] Deleted | Xiaohua Pan | 10/31/19 2:23:00 PM |

| Page 13: [15] Formatted | Xiaohua Pan | 10/31/19 2:23:00 PM |

Font: Bold, Font color: Text 1

| Page 13: [15] Formatted | Xiaohua Pan | 10/31/19 2:23:00 PM |

Font: Bold, Font color: Text 1

| Page 13: [15] Formatted | Xiaohua Pan | 10/31/19 2:23:00 PM |

Font: Bold, Font color: Text 1

| Page 13: [15] Formatted | Xiaohua Pan | 10/31/19 2:23:00 PM |

Font: Bold, Font color: Text 1

| Page 13: [16] Deleted | Xiaohua Pan | 10/22/19 1:46:00 PM |

| Page 13: [17] Deleted | Xiaohua Pan | 10/31/19 2:30:00 PM |

| Page 13: [18] Deleted | Xiaohua Pan | 10/22/19 1:56:00 PM |

| Page 13: [19] Formatted | Xiaohua Pan | 10/30/19 2:14:00 PM |

Font color: Text 1

| Page 13: [19] Formatted | Xiaohua Pan | 10/30/19 2:14:00 PM |

Font color: Text 1

| Page 13: [20] Deleted | Xiaohua Pan | 10/22/19 1:48:00 PM |

| Page 13: [20] Deleted | Xiaohua Pan | 10/22/19 1:48:00 PM |

| Page 13: [20] Deleted | Xiaohua Pan | 10/22/19 1:48:00 PM |

| Page 13: [20] Deleted | Xiaohua Pan | 10/22/19 1:48:00 PM |

| Page 13: [20] Deleted | Xiaohua Pan | 10/22/19 1:48:00 PM |

| Page 14: [21] Formatted | Xiaohua Pan | 10/30/19 2:14:00 PM |

Font color: Text 1

| Page 14: [21] Formatted | Xiaohua Pan | 10/30/19 2:14:00 PM |
|---|---|---|

Font color: Text 1

| Page 14: [21] Formatted | Xiaohua Pan | 10/30/19 2:14:00 PM |
|---|---|---|

Font color: Text 1

| Page 14: [21] Formatted | Xiaohua Pan | 10/30/19 2:14:00 PM |
|---|---|---|

Font color: Text 1

| Page 14: [22] Formatted | Xiaohua Pan | 10/30/19 2:14:00 PM |
|---|---|---|

Font color: Text 1

| Page 14: [22] Formatted | Xiaohua Pan | 10/30/19 2:14:00 PM |
|---|---|---|

Font color: Text 1

| Page 14: [23] Formatted | Xiaohua Pan | 10/30/19 2:14:00 PM |
|---|---|---|

Font color: Text 1

| Page 14: [23] Formatted | Xiaohua Pan | 10/30/19 2:14:00 PM |
|---|---|---|

Font color: Text 1

| Page 14: [24] Deleted | Xiaohua Pan | 10/25/19 10:49:00 AM |
|---|---|---|

| Page 14: [24] Deleted | Xiaohua Pan | 10/25/19 10:49:00 AM |
|---|---|---|

| Page 14: [25] Formatted | Xiaohua Pan | 10/30/19 2:14:00 PM |
|---|---|---|

Font color: Text 1

| Page 14: [25] Formatted | Xiaohua Pan | 10/30/19 2:14:00 PM |
|---|---|---|

Font color: Text 1

| Page 14: [26] Deleted | Xiaohua Pan | 10/11/19 1:37:00 PM |
|---|---|---|

| Page 14: [26] Deleted | Xiaohua Pan | 10/11/19 1:37:00 PM |
|---|---|---|

| Page 14: [26] Deleted | Xiaohua Pan | 10/11/19 1:37:00 PM |
|---|---|---|

| Page 14: [26] Deleted | Xiaohua Pan | 10/11/19 1:37:00 PM |
|---|---|---|

| Page 14: [26] Deleted | Xiaohua Pan | 10/11/19 1:37:00 PM |
|---|---|---|

| Page 14: [26] Deleted | Xiaohua Pan | 10/11/19 1:37:00 PM |
|---|---|---|

| Page 14: [26] Deleted | Xiaohua Pan | 10/11/19 1:37:00 PM |
|---|---|---|

| Page 14: [27] Deleted | Xiaohua Pan | 10/22/19 2:12:00 PM |
|---|---|---|

| Page 14: [27] Deleted | Xiaohua Pan | 10/22/19 2:12:00 PM |
|---|---|---|

| Page 14: [28] Deleted | Ichoku, Charles | 10/26/19 12:10:00 PM |
|---|---|---|

| Page 14: [28] Deleted | Ichoku, Charles | 10/26/19 12:10:00 PM |
|---|---|---|

| Page 14: [29] Deleted | Xiaohua Pan | 10/22/19 2:14:00 PM |
|---|---|---|

| Page 14: [29] Deleted | Xiaohua Pan | 10/22/19 2:14:00 PM |
|---|---|---|

| Page 14: [29] Deleted | Xiaohua Pan | 10/22/19 2:14:00 PM |
|---|---|---|

| Page 14: [30] Formatted | Xiaohua Pan | 10/30/19 2:14:00 PM |
|---|---|---|

Font color: Text 1

| Page 14: [30] Formatted | Xiaohua Pan | 10/30/19 2:14:00 PM |
|---|---|---|

Font color: Text 1

| Page 14: [31] Deleted | Xiaohua Pan | 10/28/19 2:10:00 PM |
|---|---|---|

| Page 14: [31] Deleted | Xiaohua Pan | 10/28/19 2:10:00 PM |
|---|---|---|

| Page 14: [31] Deleted | Xiaohua Pan | 10/28/19 2:10:00 PM |
|---|---|---|

| Page 14: [32] Deleted | Ichoku, Charles | 10/26/19 12:14:00 PM |
|---|---|---|

| Page 14: [32] Deleted | Ichoku, Charles | 10/26/19 12:14:00 PM |
|---|---|---|

| Page 14: [33] Deleted | Ichoku, Charles | 10/26/19 12:15:00 PM |
|---|---|---|

| Page 14: [33] Deleted | Ichoku, Charles | 10/26/19 12:15:00 PM |
|---|---|---|

| Page 14: [34] Deleted | Xiaohua Pan | 10/22/19 3:27:00 PM |
|---|---|---|

| Page 14: [34] Deleted | Xiaohua Pan | 10/22/19 3:27:00 PM |
|---|---|---|

| Page 15: [35] Deleted | Xiaohua Pan | 10/22/19 2:26:00 PM |
|---|---|---|

| Page 15: [36] Formatted | Xiaohua Pan | 10/30/19 2:14:00 PM |
|---|---|---|

Font color: Text 1, Not Highlight

| Page 15: [37] Formatted | Xiaohua Pan | 10/30/19 2:14:00 PM |
|---|---|---|

Font color: Text 1

| Page 15: [38] Deleted | Xiaohua Pan | 10/11/19 1:53:00 PM |
|---|---|---|

| Page 15: [39] Deleted | Xiaohua Pan | 10/22/19 3:52:00 PM |
|---|---|---|

| Page 15: [40] Deleted | Ichoku, Charles | 10/26/19 12:25:00 PM |
|---|---|---|

| Page 15: [41] Deleted | Xiaohua Pan | 10/22/19 3:52:00 PM |
|---|---|---|

| Page 15: [42] Formatted | Xiaohua Pan | 10/30/19 2:14:00 PM |
|---|---|---|

Font color: Text 1

| Page 15: [43] Formatted | Xiaohua Pan | 10/30/19 2:14:00 PM |
|---|---|---|

Font color: Text 1

| Page 15: [44] Formatted | Xiaohua Pan | 10/30/19 2:14:00 PM |
|---|---|---|

Font: Not Bold, No underline, Font color: Text 1

| Page 15: [45] Formatted | Xiaohua Pan | 10/30/19 2:14:00 PM |
|---|---|---|

Font color: Text 1

| Page 15: [46] Formatted | Xiaohua Pan | 10/30/19 2:14:00 PM |
|---|---|---|

Font color: Text 1

| Page 15: [47] Formatted | Xiaohua Pan | 10/30/19 2:14:00 PM |
|---|---|---|

Font color: Text 1

| Page 15: [48] Formatted | Xiaohua Pan | 10/30/19 2:14:00 PM |
|---|---|---|

Font color: Text 1

| Page 15: [49] Formatted | Xiaohua Pan | 10/30/19 2:14:00 PM |
|---|---|---|

Font color: Text 1

| Page 15: [50] Formatted | Xiaohua Pan | 10/30/19 2:14:00 PM |
|---|---|---|

Font color: Text 1

| Page 47: [51] Deleted | Xiaohua Pan | 10/30/19 2:11:00 PM |
|---|---|---|

---

## Author Response (AR2)

Dear Editor Spracklen,

We appreciate your comment and recommendation on our manuscript.
We have incorporated your suggestions and revised our manuscript. The below is the point to point responses to your two specific comments:

Your comment:

> In response to Referee #1 (Line 402) it is important to mention the uncertainties involved in evaluating aerosol emissions using AOD. This is due to the uncertainties in calculating AOD from simulated aerosol due to uncertainties in aerosol size distribution, optical properties, aerosol water uptake, vertical distribution of aerosol, meteorology etc. See Reddington et al. (2019), comparing 3 emission datasets in one model over the Amazon against both AOD and vertical aerosol profiles (https://www.atmos-chem-phys.net/19/9125/2019/acp-19-9125-2019.html). It is possible to correctly simulate the aerosol vertical profile but still underestimate AOD.

Our response:

In Sect. 4.3, we have added this paragraph "Given the sparsity of the measurements of surface and vertical concentrations at the global scale, we implemented an approach to evaluate model simulation uncertainty globally due to biomass burning aerosol emissions by evaluating the resulting AOD against those from satellite data and AERONET measurements, following the studies by Petrenko et al. (2012) and Zhang et al. (2014). We acknowledge the uncertainties in calculating AOD, such as uncertainties associated with assumptions of aerosol size distribution, optical properties, aerosol water uptake, and vertical distribution of aerosol (e.g., Reddington et al., 2019). In addition, Ge et al. (2017) showed that the choice of different meteorological fields can also lead to uncertainty in simulating the modelled aerosol loading. For instance, meteorological fields from ECMWF and National Centers for Environmental Prediction (NCEP) can yield a factor of two difference in the resulting surface $PM_{2.5}$ concentration during the fire season of September in the Maritime continents."

Your comment:

> It would also be useful to stress that not all the emission datasets include peat combustion. GFED3.1 and GFED4 do, but FINN1.5 does not. Peat combustion is an important emission source in some regions such as Equatorial Asia. For example, see: Kiely et al (2019) (https://www.atmos-chem-phys.net/19/11105/2019/acp-19-11105-2019.html). This will contribute to the differences in emissions between emission datasets in some regions.

Our response:

In Sect.4.2, we have added a new sub-section as below:

[revised manuscript text omitted]

OC biomass burning emission for 2008

(a) GFED3.1 (sum=15.65 Tg yr⁻¹)    (b) GFED4s (sum=13.76 Tg yr⁻¹)

(c) FINN1.5 (sum=19.48 Tg yr⁻¹)    (d) GFAS1.2 (sum=18.22 Tg yr⁻¹)

(e) FEER1.0 (sum=28.48 Tg yr⁻¹)    (f) QFED2.4 (sum=51.93 Tg yr⁻¹)

(Units:g m⁻² yr⁻¹)

Figure 2. The spatial distribution of annual total organic carbon (OC) biomass burning emissions
for 2008 estimated by six biomass burning emission datasets (units: g m⁻² yr⁻¹). The global annual
total amount for each dataset in 2008 is indicated in the parentheses.

[Figure]

OC biomass burning emission for 2008

|  | BONA | TENA | CEAM | NHSA | SHSA | EURO | MIDE | NHAF | SHAF | BOAS | CEAS | SEAS | EQAS | AUST |
|---|---|---|---|---|---|---|---|---|---|---|---|---|---|---|
| mean | 0.86 | 0.86 | 0.54 | 0.32 | 3.15 | 0.09 | 0.12 | 4.07 | 6.18 | 3.90 | 1.52 | 1.81 | 0.39 | 0.77 |
| std | 0.78 | 1.15 | 0.45 | 0.20 | 1.69 | 0.08 | 0.19 | 1.81 | 3.29 | 2.65 | 1.16 | 1.47 | 0.18 | 0.59 |
| max/min | 10.12 | 17.15 | 10.04 | 5.57 | 4.17 | 12.65 | 66.44 | 2.89 | 3.10 | 5.28 | 7.88 | 7.27 | 3.72 | 14.77 |
| cv | 0.90 | 1.33 | 0.85 | 0.64 | 0.54 | 0.89 | 1.61 | 0.44 | 0.53 | 0.68 | 0.76 | 0.81 | 0.46 | 0.77 |
| rank of cv | 12 | 13 | 10 | 5 | 4 | 11 | 14 | 1 | 3 | 6 | 7 | 9 | 2 | 8 |

Legend    GLOB_TOT

GFED3.1    15.65
GFED4s    13.76
FINN1.5    19.48
GFAS1.2    18.22
FEER1.0    28.48
QFED2.4    51.93

[revised manuscript text omitted]

**BC biomass burning emission for 2008**

[Figure]

[Figure]

¶
Page Break
¶

Figure S1. The spatial distribution of annual total black carbon (BC) biomass burning emissions for 2008 estimated by six biomass burning emission datasets. The global total amount is indicated in the parentheses. The fourteen selected AERONET sites are labeled as magenta dots.

BC biomass burning emission for 2008

| | BONA | TENA | CEAM | NHSA | SHSA | EURO | MIDE | NHAF | SHAF | BOAS | CEAS | SEAS | EQAS | AUST |
|---|---|---|---|---|---|---|---|---|---|---|---|---|---|---|
| mean | 0.05 | 0.07 | 0.06 | 0.04 | 0.39 | 0.01 | 0.01 | 0.55 | 0.80 | 0.26 | 0.16 | 0.18 | 0.04 | 0.10 |
| std | 0.05 | 0.08 | 0.04 | 0.03 | 0.22 | 0.01 | 0.02 | 0.28 | 0.46 | 0.19 | 0.11 | 0.12 | 0.02 | 0.08 |
| max/min | 18.45 | 16.54 | 8.05 | 5.77 | 4.18 | 11.83 | 57.96 | 3.54 | 3.97 | 6.48 | 7.00 | 4.98 | 3.56 | 18.95 |
| cv | 1.00 | 1.22 | 0.70 | 0.66 | 0.55 | 0.78 | 1.53 | 0.50 | 0.57 | 0.73 | 0.71 | 0.66 | 0.49 | 0.78 |
| rank of cv | 12 | 13 | 7 | 6 | 3 | 11 | 14 | 2 | 4 | 9 | 8 | 5 | 1 | 10 |

[Figure]

Legend — GLOB_TOT
GFED3.1 1.76
GFED4s 1.65
FINN1.5 1.83
GFAS1.2 1.99
FEER1.0 3.66
QFED2.4 5.54

(Units: Tg yr$^{-1}$)

Figure S2. The regional annual total black carbon (BC) biomass burning emissions for 2008 in six biomass burning emission datasets in 14 regions (units: Tg yr$^{-1}$). The global annual total amount is listed after the name of each dataset (GLOB_TOT). Relevant statistics for the six BB emission datasets in each region are also listed at the top of the panel in blue under the short name of each region, with the mean of the six BB emission datasets in the first row. Three different methods to measure the dispersion of the six BB emission datasets are shown as well: one absolute method, i.e., the standard deviation (std) in the second row, and two relative methods, i.e., the ratio of max (maximum) to min (minimum) shown in the third row, and the coefficient of variation (cv), defined as the ratio of the std to the mean, in the fourth row. The rankings of the regions reflecting the dispersion of the BB emissions datasets according to cv are shown in the fifth row (i.e., a ranking of 1 means that this region shows the least spread among the six BB emissions datasets, while a ranking of 14 indicates this region has the largest spread in the 14 regions).

[Figure]

Monthly variation of BC biomass burning emission for 2008

Figure S3. Monthly variation of black carbon (BC) biomass burning emissions for 2008 in six
biomass burning emission datasets in 14 regions and the global (GLOB, highlighted with a black
box). The annual total emission is listed in the right side of each panel.

**Table S1.** The regionally-averaged monthly mean AOD from MISR along with model biases (i.e., model minus MISR) in seven model
experiments, i.e., NOBB, GFED3.1, GFED4s, FINN1.5, GFAS1.2, FEER1.0, and QFED2.4, for September and April 2008.

| Dataset | BONA | TENA | CEAM | NHSA | SHSA | EURO | MIDE | NHAF | SHAF | BOAS | CEAS | SEAS | EQAS | AUST | GLOB |
|---------|------|------|------|------|------|------|------|------|------|------|------|------|------|------|------|
| **September 2008** | | | | | | | | | | | | | | | |
| **MISR** | 0.127 | 0.129 | 0.133 | 0.141 | 0.188 | 0.139 | 0.334 | 0.411 | 0.331 | 0.135 | 0.219 | 0.257 | 0.138 | 0.085 | 0.218 |
| **NOBB** | 0.04 | 0.011 | -0.031* | -0.083 | -0.132 | 0.084 | -0.013 | -0.05 | -0.283 | 0.043 | 0.053 | 0.025 | -0.047 | -0.031 | -0.029 |
| **GFED3.1** | 0.042 | 0.013 | -0.031 | -0.074 | -0.078 | 0.085 | -0.012 | -0.046 | -0.178 | 0.048 | 0.055 | 0.026 | -0.039 | -0.02 | -0.012 |
| **GFED4s** | 0.044 | 0.014 | -0.031 | -0.075 | -0.081 | 0.085 | -0.012 | -0.047 | -0.208 | 0.047 | 0.056 | 0.025 | -0.04 | -0.023 | -0.015 |
| **FINN1.5** | 0.041 | 0.013 | -0.031 | -0.07 | -0.04 | 0.08 | -0.013 | -0.047 | -0.21 | 0.045 | 0.05 | 0.016 | -0.033 | -0.025 | -0.012 |
| **GFAS1.2** | 0.042 | 0.013 | -0.031 | -0.074 | -0.08 | 0.08 | -0.013 | -0.047 | -0.208 | 0.046 | 0.05 | 0.016 | -0.029 | -0.02 | -0.016 |
| **FEER1.0** | 0.044 | 0.016 | -0.03 | -0.061 | -0.021 | 0.088 | -0.009 | -0.039 | -0.079 | 0.049 | 0.061 | 0.027 | -0.032 | -0.01 | 0.006 |
| **QFED2.4** | 0.058 | 0.04 | -0.023 | -0.058 | 0.02 | 0.097 | -0.004 | -0.035 | -0.044 | 0.058 | 0.068 | 0.028 | -0.029 | -0.002 | 0.02 |
| **April 2008** | | | | | | | | | | | | | | | |
| **MISR** | 0.192 | 0.16 | 0.182 | 0.207 | 0.067 | 0.148 | 0.381 | 0.446 | 0.096 | 0.221 | 0.324 | 0.363 | 0.118 | 0.049 | 0.242 |
| **NOBB** | -0.041 | -0.016 | -0.083 | -0.086 | -0.016 | 0.077 | -0.059 | -0.082 | -0.042 | -0.005 | 0.017 | -0.074 | -0.028 | -0.004 | -0.029 |
| **GFED3.1** | -0.023 | -0.008 | -0.077 | -0.082 | -0.014 | 0.079 | -0.058 | -0.078 | -0.04 | 0.104 | 0.055 | -0.057 | -0.027 | -0.002 | -0.015 |
| **GFED4s** | -0.026 | -0.008 | -0.076 | -0.081 | -0.015 | 0.081 | -0.057 | -0.079 | -0.039 | 0.095 | 0.05 | -0.063 | -0.027 | -0.002 | -0.016 |
| **FINN1.5** | -0.031 | -0.006 | -0.056 | -0.072 | -0.013 | 0.075 | -0.055 | -0.068 | -0.037 | 0.041 | 0.069 | 0.014 | -0.024 | -0.002 | -0.008 |
| **GFAS1.2** | -0.014 | -0.005 | -0.07 | -0.075 | -0.014 | 0.073 | -0.057 | -0.073 | -0.039 | 0.164 | 0.07 | -0.058 | -0.025 | -0.001 | -0.008 |
| **FEER1.0** | -0.017 | 0.003 | -0.054 | -0.059 | -0.011 | 0.089 | -0.053 | -0.068 | -0.036 | 0.162 | 0.083 | -0.028 | -0.024 | 0 | -0.001 |
| **QFED2.4** | 0.018 | 0.027 | -0.026 | -0.054 | -0.008 | 0.099 | -0.047 | -0.059 | -0.032 | 0.329 | 0.17 | 0.038 | -0.022 | 0.004 | 0.031 |

*Highlighted in blue if negative bias.*